# Structure and efflux mechanism of the yeast pleiotropic drug resistance transporter Pdr5

Andrzej Harris [1,10], Manuel Wagner[2,8,10], Dijun Du[1,9,10], Stefanie Raschka[2], Lea-Marie Nentwig[2], Holger Gohlke [3,4,5,6], Sander H. J. Smits [2,7], Ben F. Luisi [1✉] & Lutz Schmitt [2✉]

Pdr5, a member of the extensive ABC transporter superfamily, is representative of a clinically relevant subgroup involved in pleiotropic drug resistance. Pdr5 and its homologues drive drug efflux through uncoupled hydrolysis of nucleotides, enabling organisms such as baker's yeast and pathogenic fungi to survive in the presence of chemically diverse antifungal agents. Here, we present the molecular structure of Pdr5 solved with single particle cryo-EM, revealing details of an ATP-driven conformational cycle, which mechanically drives drug translocation through an amphipathic channel, and a clamping switch within a conserved linker loop that acts as a nucleotide sensor. One half of the transporter remains nearly invariant throughout the cycle, while its partner undergoes changes that are transmitted across inter-domain interfaces to support a peristaltic motion of the pumped molecule. The efflux model proposed here rationalises the pleiotropic impact of Pdr5 and opens new avenues for the development of effective antifungal compounds.

[1] Department of Biochemistry, University of Cambridge, Cambridge, UK. [2] Institute of Biochemistry, Heinrich Heine University Düsseldorf, Universitätsstraße 1, Düsseldorf, Germany. [3] Institute of Pharmaceutical and Medicinal Pharmacy, Heinrich Heine University Düsseldorf, Universitätsstraße 1, Düsseldorf, Germany. [4] John von Neumann Institute for Computing (NIC), Jülich Supercomputing Centre (JSC), Forschungszentrum Jülich GmbH, Jülich, Germany. [5] Institute of Biological Information Processing (IBI-7: Structural Biochemistry), Forschungszentrum Jülich GmbH, Jülich, Germany. [6] Institute of Bio- and Geosciences (IBG-4: Bioinformatics), Forschungszentrum Jülich GmbH, Jülich, Germany. [7] Center for Structural Studies, Heinrich Heine University Düsseldorf, Universitätsstraße 1, Düsseldorf, Germany. [8] Present address: Medac GmbH, Theatherstraße 6, Wedel, Germany. [9] Present address: School of Life Sciences and Technology, ShanghaiTech University, Pudong, Shanghai, China. [10] These authors contributed equally: Andrzej Harris, Manuel Wagner, Dijun Du. ✉email: bfl20@cam.ac.uk; lutz.schmitt@hhu.de

The ATP-binding cassette (ABC) transporters comprise an extensive family of membrane transport proteins. These machines, found in all domains of life, are the primary source of active transport across the cell membrane and share a common architecture of two pairs of domains: the transmembrane (TMD) and the nucleotide-binding domain (NBD)[1]. ATP binding and hydrolysis by the NBDs energises the transport cycle by driving conformational changes in the TMDs that enable substrate passage across the bilayer.

ABC transporters recognise a wide range of substrates, including nutrients and cell-wall components[2,3]. Many of the transporters (especially in eukaryotes) export toxic compounds from cells; indeed, overexpression of these proteins is a key contributor to multi-drug resistance (MDR) phenotypes. In plant and fungi, several efflux pumps involved in MDR are found within the pleiotropic drug resistance (PDR) subfamily, whose members confer resistance to structurally and functionally unrelated drugs and xenobiotics[2]. Pdr5 is the founding member of a full-length transporter subfamily within the G subfamily of ABC transporters, which otherwise contains half-size transporters. The distinguishing features of the PDR transporters include a structural repeat in which the NBD domain is at the N-terminus of both pseudo-protomers, resulting in a reverse transmembrane topology in comparison with other ABC transporters and similar to half-size ABCG transporters. PDR proteins also harbour a characteristic N-terminal extension of approximately 160–170 amino acid residues of unknown function[3].

Since its discovery 30 years ago[4], the ABC transporter Pdr5 from *Saccharomyces cerevisiae* has become an established and widely studied model for PDR proteins in fungi that include major pathogens[5]. Whilst its physiological substrate (or substrates) is not known, Pdr5 was shown to transport a wide variety of chemicals, including azoles, ionophores, antibiotics and other xenobiotics[6,7]. Pdr5 homologues in *Candida albicans* contribute to increased mortality in immune-compromised patients[8,9] and fungicide resistance of the plant pathogen *Botrytis cinerea*[5,10].

ABC transporters share several signature motifs in the NBDs[11]. Both NBDs contribute residues of these conserved motifs to form two nucleotide-binding sites (NBSs) that bind and hydrolyse ATP. In Pdr5, one of the NBS is catalytically active (NBS2), while the other (NBS1) is inactive due to multiple substitutions in crucial residues of all but one motif. While this asymmetry is shared with many other ABC transporters (e.g., CFTR)[12], the PDR subfamily represents its most extreme case[13]. It is unclear how ATPase-deficiency at this nucleotide-binding site supports the transport process.

Despite three decades of extensive study of Pdr5 biochemistry, the protein has evaded structural characterisation, leaving a wealth of transport data mostly untapped. Taking advantage of recent improvements in homogenous preparations of Pdr5[14] (Supplementary Fig. 1) and advances in single-particle electron cryo-microscopy (cryo-EM) of membrane proteins[15], we set about to study the molecular architecture of the transporter and obtain insight into its distinctive mechanism.

In this work, we report the cryo-EM structure of Pdr5 from *S. cerevisiae*, providing details of the fold and conformation of a representative of the extensive PDR subfamily. One distinguishing feature revealed by our model is a conserved linker region located near the non-catalytic NBS1 site, which acts as a nucleotide sensor. Additionally, through our high-resolution structures of the protein with a bound transport ligand in substrate-receiving, and in substrate-releasing conformation, we visualise the translocation of drugs from the cytoplasm to the cell exterior. Based on these structural data and earlier functional studies, we propose a model for asymmetric action and drug efflux in Pdr5.

## Results and discussion

**Pdr5 in detergent-free system can be imaged with high-resolution cryo-EM.** The membrane environment is critical for the activity of highly allosteric transporters like Pdr5. Several purification protocols have been developed that sustain membrane proteins in a native membrane-like environment[16–19]. In this study, we reconstituted Pdr5 in peptidiscs, which are short amphipathic bi-helical peptides[17] compatible with single-particle cryo-EM[20,21], after purification from *S. cerevisiae* cell membranes. Reconstituted Pdr5 retained ATPase activity (Supplementary Fig. 2). Compared to the detergent-solubilised state, the kinetic parameter $V_{max}$ (72.8 ± 2.9 nmol min$^{-1}$ mg$^{-1}$) decreased 3-fold and $K_M$ (0.65 ± 0.01 mM) increased 1.5-fold One possible explanation is that the peptidisc belt, more rigid than the detergent shell, interfered with the efficiency of ATP hydrolysis. This preparation yielded homogenous particles readily visualised by electron cryo-microscopy (Supplementary Fig. 2a, b). We collected four datasets: apo Pdr5, Pdr5 with added ATP, Pdr5 with added ATP and sodium orthovanadate (V$_i$), and Pdr5 with ATP and transport substrate rhodamine 6 G (R6G)[7,22,23]. V$_i$ is an analogue of inorganic phosphate, and a potent inhibitor of ATPases that traps ABC transporters in a transition-state conformation by mimicking the action of the γ-phosphate during ATP hydrolysis[24–27].

The 3D reconstruction of Pdr5 was carried out predominantly in RELION[28] (Supplementary Fig. 4) and yielded four near-atomic resolution cryo-EM maps. The sample with Pdr5 alone produced a map containing no nucleotide ligands, which we refer to as the apo-Pdr5 state (Supplementary Fig. 3d). The Pdr5 sample with added ATP yielded a map containing the hydrolysed nucleotide ADP in the canonical and ATP in the inactive site, called here ADP-Pdr5 (Supplementary Fig. 3f). The sample with added ATP and R6G produced a map with the nucleotide composition as above, but also with a bound substrate in the transport cavity, called here R6G-Pdr5 (Supplementary Fig. 3i). The orthovanadate dataset contained Pdr5 in a distinct state, showing the vanadate-trapped hydrolytic intermediate and termed AOV-Pdr5, with ATP in the inactive and ADP-VO$_4^{3-}$ (or V$_i$) in the canonical site (Supplementary Fig. 3h). These maps represent the two most distinct conformational states of Pdr5: inward-facing (substrate-receiving) and outward-facing (substrate-releasing).

The resolution of the maps ranged between 2.9 Å and 3.8 Å, calculated using Fourier shell correlation[29] (Supplementary Fig. 3d, f, h, j and Supplementary Table 1), revealing well-resolved side-chains of bulkier amino acid residues (Fig. 1e–f). The quality of the maps enabled the majority (90% amino acid residues) of the structure of this 170-kDa transporter to be traced with certainty for all four states. Some of the external loops of the NBD domains had unresolved features, most likely due to flexibility (Supplementary Figs. 5, 6).

**Pdr5 adopts the architecture of an asymmetric, full-size ABC transporter.** Pdr5 clearly adopts the fold of type II exporters[30] or, according to the newly suggested nomenclature for ABC transporters, a type V fold[31]. The Pdr5 model (Fig. 1b) reveals a pseudo-dimeric architecture, undoubtedly the result of gene duplication[5]. The NBD domains of Pdr5 are structurally similar to those of other members of the superfamily and comprise the RecA-like[32,33] and helical sub-domain (the bottom and top halves of NBD1 in Fig. 1c). The NBD1 and NBD2 of Pdr5 are structurally similar (Fig. 1c), sharing 27% sequence identity. NBD2 is the more conserved of the two; together with the C-loop of NBD1, it forms the composite canonical hydrolytic NBS2,

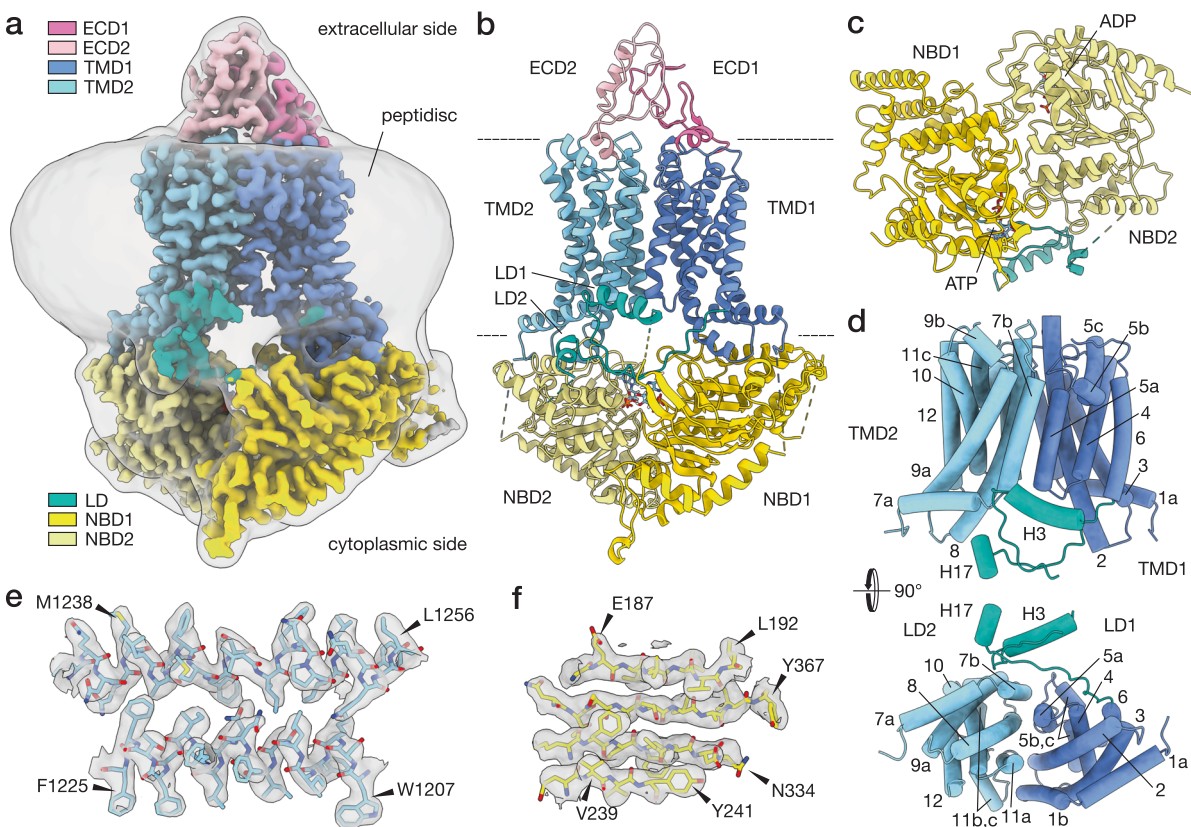

**Fig. 1 Structural features of Pdr5. a** Depicted here is the cryo-EM map of Pdr5 in inward-open conformation (ADP-Pdr5). The map is coloured according to the domain organisation (see key). The grey transparent envelope surrounding the protein is a Gaussian-filtered (2σ) version of the same map, showing the extent and shape of the peptidisc shell around Pdr5. **b** Atomic model of Pdr5 built and refined against the map, in cartoon representation. **c** Top view (cytoplasmic side) of the nucleotide-binding domains of Pdr5, showing the position of the bound nucleotides. The linker domain is also depicted. **d** Side and top views of the transmembrane domain of Pdr5 with α-helices shown schematically as cylinders and numbered sequentially (TH1a–12) from the N-terminus (cf. Supplementary Fig. 5 and S6). This panel also includes the linker domain and its two helices H3 and H17. **e**, **f** Two portions of the ADP-Pdr5 cryo-EM map, showing the reconstructed volume around representative secondary structural features: **e** α-helices TH7b and TH8 of TMD1 domain and (**f**) a β-sheet in NBD1 domain. Arrowheads point to amino acid residues that have been arbitrarily chosen for orientation purposes. Abbreviations: ECD, extracellular domain; LD, linker domain; NBD, nucleotide-binding domain; TMD, transmembrane domain.

whereas NBD1 and the C-loop of NBD2 cannot sustain ATP hydrolysis as its NBS1 is degenerated[13,34].

As with the NBDs, the TMDs of Pdr5 are pseudo-dimeric (Fig. 1d). Each TMD contains five long α-helices that span the lipid bilayer, a broken transmembrane α-helix that makes a near-90° bend inside the membrane (TH5 in TMD1 and TH11 in TMD2), and an N-terminal amphipathic α-helix that lies on the membrane surface (TH1a and 7a) (Fig. 1d and Supplementary Fig. 6). The latter two are also present in ABCG2[35] and ABCG5/G8[30], the closest structurally characterised relatives of Pdr5 (Supplementary Table 2 and Fig. 2). We also compared Pdr5 with other members of the ABCG subfamily, found in bacteria: Wzm-Wzt[36] and TarGH[37]. As shown in Fig. 2, the structural differences are more prominent with the latter two proteins (see also: Supplementary Table 2) emphasising that Pdr5 is more closely related to the human ABCG subfamily. The space between TMD1 and TMD2 is the location of the substrate channel in ABC transporters[1,38]. Two TMD α-helices are of particular importance to transport: the so-called coupling helix (CpH) and connecting helix (CnH)[39]. In Pdr5, the CpHs are in the cytoplasmic portion of TH2 and TH8. The CnHs are the amphipathic helices TH1a and TH7a of TMD1 and TMD2 (Fig. 1d and Supplementary Fig. 6). These helices form the main site of the interaction between the TMD and the NBD. In ABCG2, these are also pivot points for the structural transition between the substrate-bound

(or nucleotide-free) and substrate-releasing (ATP-bound) conformations of the transporter[35,39].

**Pdr5 contains a nucleotide-sensing linker domain.** One distinctive feature of Pdr5 is the linker domain (LD), situated between the two halves of Pdr5 and composed of two distinct stretches. One part (LD1) is made from a loop extrusion of 30 amino acid residues situated between the first two β-sheet strands of NBD1; it also contains an amphipathic α-helix (H3) that rests on the surface of the lipid bilayer (Fig. 1b–d, Supplementary Fig. 5 and Supplementary Fig. 6). The other part (LD2) is formed by a polypeptide chain that links the C-terminus of TMD1 with the N-terminus of NBD2 and is shaped like an arch that rises from the surface of the bilayer, goes past the first stretch of the LD and ends with a short α-helical fragment (H17) that is almost perpendicular to the long helix in the domain. The 44 residues connecting the short helix H17 with the NBD2 could not be resolved, due to flexibility (Fig. 1b–d, Supplementary Figs. 5, 6). The base of the loop of the first stretch and the middle part of the arch in the second stretch directly contact the degenerated ATP-binding site in NBD1 (Figs. 1b, 3). In the apo-Pdr5, the arch of the LD2 connecting the TMD1 and NBD2 is not as well defined in the map, and the short helix H17 could not be resolved (Fig. 3). This could indicate that ATP binding in the degenerated site has a

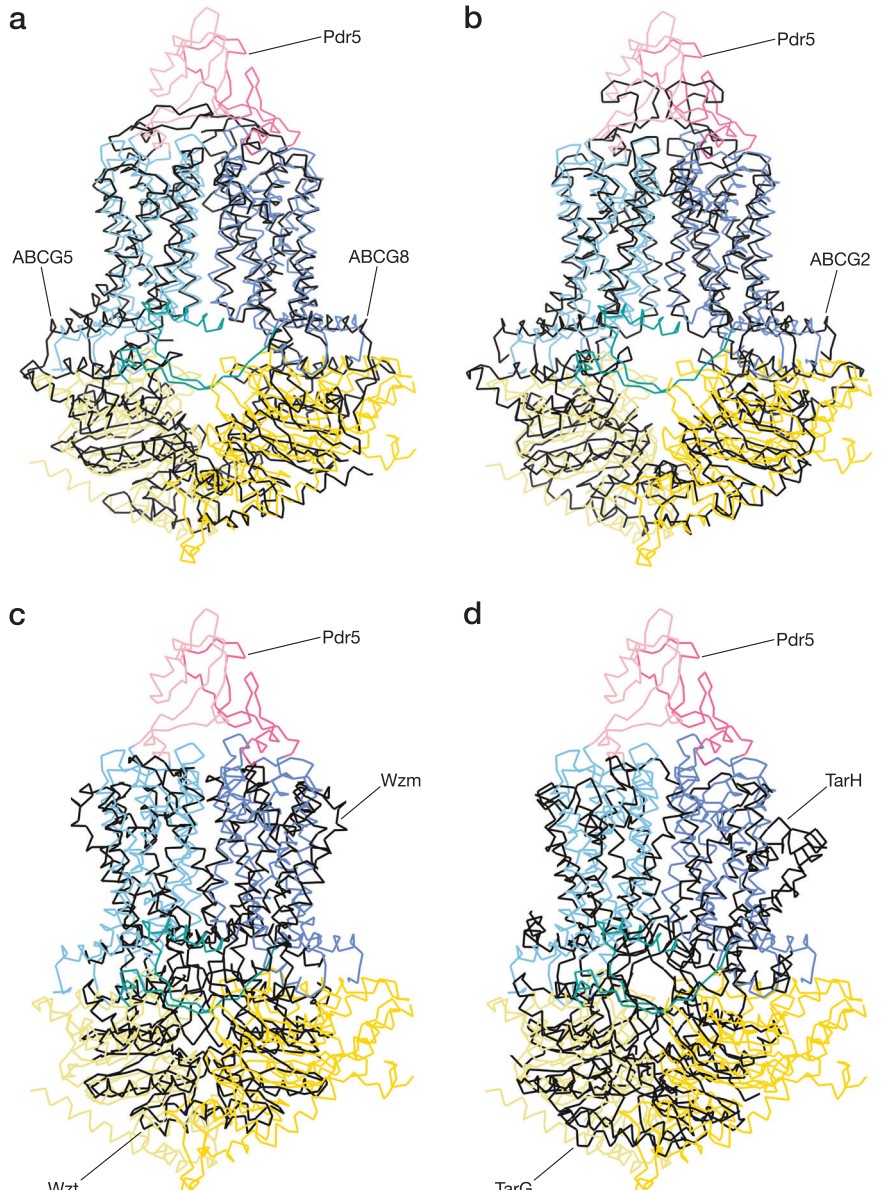

**Fig. 2 Structural comparison of Pdr5 and related transporters.** The panels show the superposition of Pdr5 and four related ABC transporters: **a** human sterol transporter ABCG5/8 (PDB-ID: 5DO7), **b** human multi-drug transporter ABCG2 (PDB-ID: 6XVI), **c** bacterial O-antigen polysaccharide transporter Wzm-Wzt (PDB-ID: 7K2T), **d** bacterial teichoic acid transporter TarGH (PDB-ID: 6JBH). The r.m.s.d. (root-mean-square deviations) of atomic positions in the amino-acid backbone are, for ABCG5/8: 4.3 Å, ABCG2: 3.9 Å, Wzm-Wzt: 9.2 Å, TarGHL: 14 Å. See also: Supplementary Table 2.

stabilising effect on that part of the LD, possibly through the contact of E804 with the ribose, as well as an electrostatic interaction of the Q801 side-chain with the triphosphate (at least in ADP-Pdr5) (Fig. 3 and Supplementary Fig. 10). The unresolved region of Pdr5 between the H17 helix and NBD2 contains a ubiquitination site on K825[40,41], but there is no indication yet of other complex interactions in that part of the transporter.

Sequence alignment of the PDR family revealed that the linker region contacting ATP contains a motif, highly conserved in the PDR subfamily, with a consensus sequence MQKGEIL (Fig. 3) and the glutamine corresponding to Q801 mentioned above. The most highly conserved (95%) residue in the motif is the glutamate (E804 in Pdr5); in all other cases, this residue is an aspartate. In our structure, E804 contacts N1011 and, together with the two hydrophobic residues I805 and L806, positions the arch of LD2 between the loop extension of LD1 and NBD2 (Fig. 3). In the apo-Pdr5 structure, the residues of the conserved MQKGEIL

motif cannot be discerned in the density, indicating that the motif plays a structural role, perhaps ensuring that the linker domain becomes more conformationally rigid when a nucleotide is bound in NBS1.

We sought to analyse the motif biochemically, through a mutation study. We targeted residues Q801 to I805, changing the sizes of the side-chains, their polarity, or charge (Supplementary Fig. 7). Firstly, we assayed the resistance of these mutants against four drugs (rhodamine 6G, ketoconazole, fluconazole, and cycloheximide) (Supplementary Fig. 7). Certain mutants displayed increased sensitivity, albeit in a substrate-dependent manner (Supplementary Table 3). Highly significant increases in sensitivity were observed for Q801 mutations to A or V for cycloheximide, simultaneously not showing any influence in the case of ketoconazole and rhodamine 6G, with some effect for the valine mutation and fluconazole. For E804, mutation to A showed a highly significant effect only in the case of fluconazole. Overall,

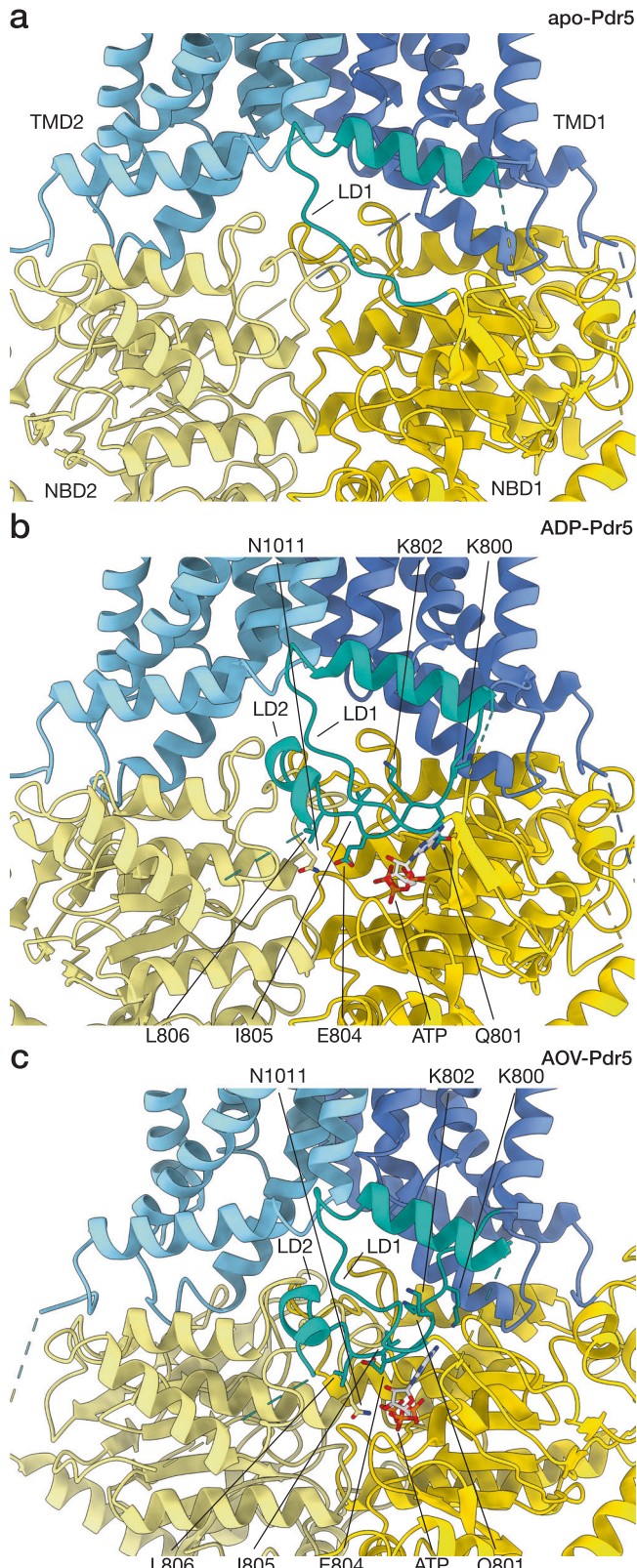

**Fig. 3 Structure of Pdr5 linker domain.** Section of the Pdr5 structure surrounding the linker domain is shown in cartoon representation. ATP in the non-hydrolytic site and some of the conserved residues of the linker domain are depicted in stick representation. The three panels represent different states of Pdr5: (**a**) the inward-facing apo-Pdr5, (**b**) the inward-facing ADP-Pdr5, (**c**) the outward-facing AOV-Pdr5. Abbreviations: LD, linker domain; NBD, nucleotide-binding domain; TMD, transmembrane domain.

retention of native lipid environment. To account for differences in expression level or effective membrane insertion, Coomassie-stained SDS-PAGE (Supplementary Fig. 8a) was analysed by densitometry. To enable direct comparison, signal intensity of Pma1 (Plasma membrane ATPase 1) was used as internal control. Based on this analysis, the difference between the expression of Pdr5 wild-type and the mutants was below 10%, and thus it was not taken into account in the quantitative analysis of the ATPase activity (Supplementary Fig. 8b). Supplementary Table 4 summarises the statistical analysis of the kinetic parameters. Whilst $K_M$ values of the wild-type and the mutants were not significantly different, the $V_{max}$ values of all mutants except I805S were highly significantly ($p < 0.0001$) decreased. This suggests that the highly conserved MQKGEIL motif not only senses the nucleotide in the degenerated NBS1, but is also involved in communication with the active NBS1. This scenario is reminiscent of the case of converting the degenerate site of Pdr5 into an active site that completely abolished ATPase activity[13]. Furthermore, it highlights that Pdr5 is an uncoupled transporter, and that even at the reduced $V_{max}$ value, it is still capable of conferring drug resistance.

**Pdr5 contains a structurally unique extracellular domain.** The extracellular (or apical) domain of Pdr5 (labelled as ECD1 and ECD2 on Fig. 1b) interrupts the TMDs just before their C-terminal α-helices: TH6 and TH12 (Fig. 1b and Supplementary Fig. 6). Most of the ECD is composed of structured loops, with two short helices (ECD1-H16 and ECD2-H29) at the base of the domain, and a slightly longer, distorted helix facing the extracellular milieu (ECD2-H28). Each half of the ECDs contains disulfide bridges (C722 and C742 in ECD1, and C1411 and C1455 as well as C1427 and C1452 of ECD2), which are probably involved in the stabilisation of the local fold. Interestingly, there is no inter-domain disulfide bridge like that seen in ABCG2[35]. In our cryo-EM map, the backbone of the asparagine N734 in ECD1 is well resolved, and its side-chain has additional density that is likely due to glycosylation (Supplementary Fig. 9), consistent with mass spectrometry profiling[42].

**NBS1 of Pdr5 does not support ATP hydrolysis.** A number of important ABC transporters possess atypical NBS that have greatly diminished or no hydrolytic abilities due to substitutions of key residues in canonical motifs. In *Saccharomyces* sp. the three major drug efflux pumps (Yor1, Snq2 and Pdr5) have an atypical NBS alongside the canonical one. Endowing NBS1 of Pdr5 with hydrolytic capabilities abolishes substrate transport[13].

The non-catalytic site of Pdr5 differs from the canonical NBS2 in all but one of the conserved regions: the D-loop. The signature motif of the PDR family of ABC transporters, VSGGE, is present in NBD1, and correctly supplying the relevant amino acid residues to perform hydrolysis in NBS2, when the domains come together, as part of the catalytic cycle of ABC transporters. Vice versa, the signature motif of NBD2, LNVEQ, is mutated completely and does not complement NBS1 (Fig. 4). However, perhaps the most consequential mutation is found in the Walker

the drug resistance analysis of this conserved motif supports its role as a crucial structural element in Pdr5, and the PDR family. However, it also revealed a rather complex pattern that highly depends on the nature of the drug.

The ATPase activity of these mutants was also analysed, using plasma membrane vesicles (Supplementary Fig. 8). Pdr5-enriched plasma membranes were selected over purified protein to ensure

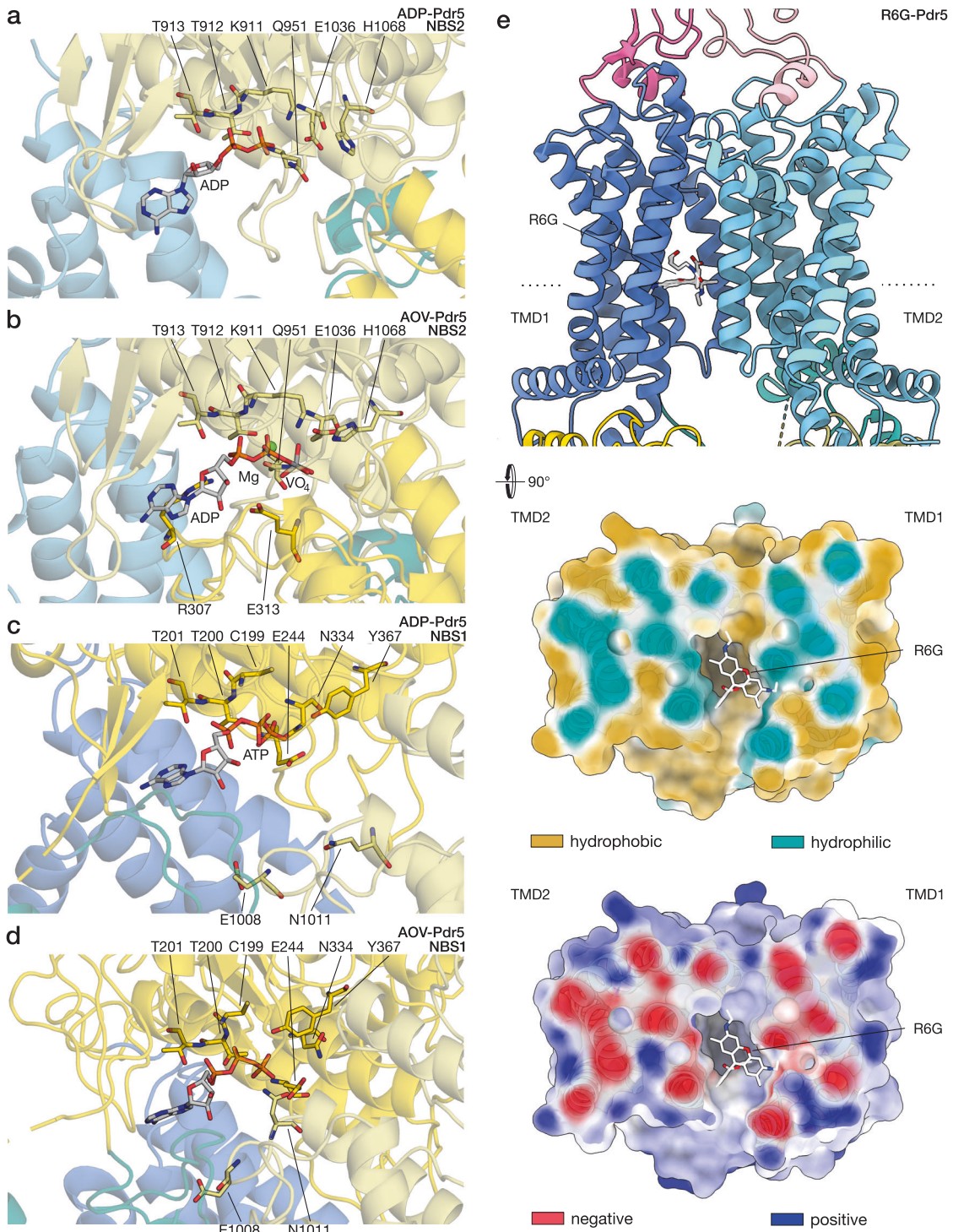

**Fig. 4 Nucleotide and substrate binding to Pdr5.** The panels illustrate the contacts between Pdr5 with (**a**–**d**) nucleotides and (**e**) rhodamine 6 G. Some of the residues that are important for nucleotide binding and/or catalysis are highlighted and labelled (cf. Supplementary Fig. 10). The domain colouring follows the scheme of the previous figure. **a** The catalytic site (NBS2) of Pdr5 in inward-facing conformation (ADP-Pdr5) with bound ADP. **b** The catalytic site of outward-facing Pdr5 (AOV-Pdr5) in a vanadate-trapped state that mimics the intermediate step of hydrolysis with an ADP-orthovanadate molecule. A co-ordinated magnesium ion is depicted as a green sphere. **c** The deviant, inactive site (NBS1) of Pdr5 in the inward-facing conformation (ADP-Pdr5), showing the bound ATP. **d** The same site in the outward-facing conformation (AOV-Pdr5), The residues surrounding the nucleotide are at the corresponding positions of the catalytic site. **e** Side view and cross-section of the transmembrane part of Pdr5 in inward-facing conformation (Pdr5-R6G). Dotted line on the side-view denotes the position of the cross-section. The efflux substrate rhodamine 6G is bound in the entrance cavity between TMD1 and TMD2 . The cross-section of Pdr5 is coloured by hydrophobicity, from turquoise (hydrophilic) to tan (most hydrophobic), and electrostatic potential, from red (negative) to blue (positive). Abbreviations: AOV, ADP-orthovanadate; R6G, rhodamine 6G; TMD, transmembrane domain.

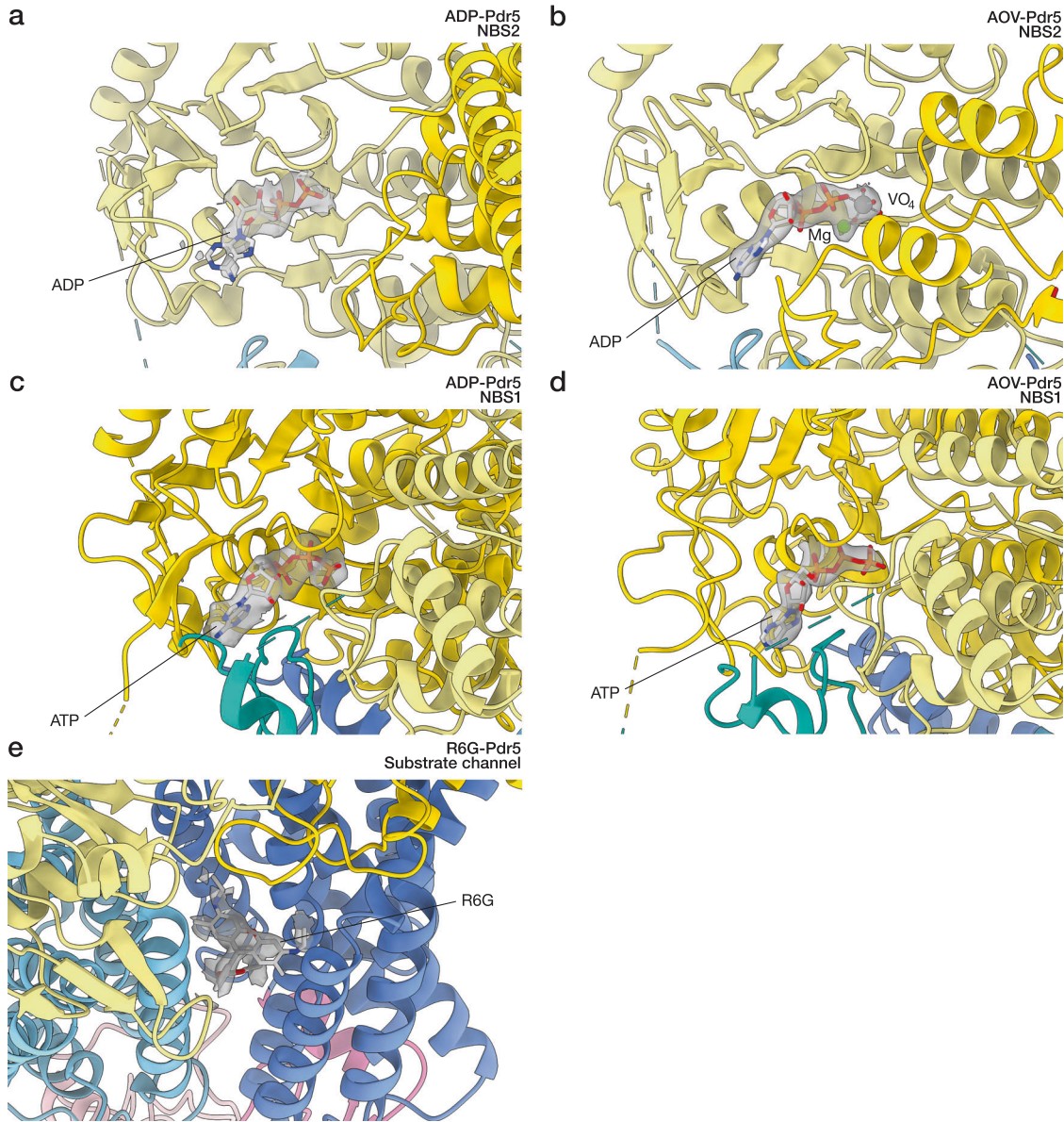

**Fig. 5 Cryo-EM maps around Pdr5 ligands.** The panels show the details of the cryo-EM map surrounding (**a**–**d**) the nucleotides and (**e**) rhodamine 6G (cf. Fig. 4 and Supplementary Fig. 10). The domain colouring follows the scheme of the previous figures. **a** The catalytic site (NBS2) of Pdr5 in inward-facing conformation (ADP-Pdr5) with bound ADP. **b** The catalytic site of outward-facing Pdr5 (AOV-Pdr5) in a vanadate-trapped state that mimics the intermediate step of hydrolysis with an ADP-orthovanadate molecule. Vanadium is shown as a grey sphere and co-ordinated magnesium ion is depicted as a green sphere. **c** The deviant, inactive site (NBS1) of Pdr5 in the inward-facing conformation (ADP-Pdr5), showing the bound ATP. **d** The same site in the outward-facing conformation (AOV-Pdr5).

B motif of the deviant ATP-binding site. Where the canonical site has a glutamate residue (E1036), the deviant has an asparagine (N334) in the same position (Fig. 4c), with a tremendous impact on its ATPase activity. As the glutamate of Walker B polarises a water nucleophile in the catalytic site[22], the hydrolysis of ATP cannot be accelerated with the N334 substitution. In the canonical sites of full-length transporters, a similar inhibitory effect has been achieved by mutating the catalytic glutamate to glutamine, which resembles the protonated form of glutamate (formed upon polarising the water)[43–45]. The non-catalytic ATP-binding site does not show the presence of a magnesium ion, although in the case of AOV-Pdr5, the map resolution in the region may prevent unambiguous assignment (Figs. 4c, 5).

**NBS2 of Pdr5 is a canonical catalytic site of an ABC transporter**. The other nucleotide-binding site, NBS2, is a functional catalytic centre, capable of ATP hydrolysis and endowed with the canonical motifs that support enzymatic function[5,13,22], including the D-loop, which provides a crucial point of contact between the two halves of the NBD dimer, coupling ATP hydrolysis, NBD dimerisation, and substrate transport[1,46].

Our structures of Pdr5 provide a glimpse into the catalytic site of the transporter in two forms: with bound ADP (ADP-Pdr5) and with an ADP-orthovanadate complex (AOV-Pdr5) (Fig. 4a–b and Fig. 5a–b, respectively) in NBS2. The ADP-Pdr5 structure was solved using Pdr5 samples with added ATP, indicating that the majority of ATP was hydrolysed before flash-freezing in

liquid ethane. It also suggests that in the post-hydrolysis stage ADP remains bound to the catalytic site, making ADP-Pdr5 the default resting state of the transporter in vivo. We were able to mimic the hydrolysis transition state of the transporter using orthovanadate. The nucleotides bind the catalytic pocket in a manner similar to other ABC transporters, reflecting the high conservation of this site[27,35,39]. Among residues that contact the nucleotide found in characteristic motifs, of particular importance are: K911, T912 and T913 in Walker A, with the lysine residue binding the terminal γ-phosphate of ATP; Q951 of the Q-loop, co-ordinating the magnesium ion in the ADP-$V_i$ complex; E1036 of Walker B, mentioned before; H1068 of the H-loop, which acts as a proton-shuttle in the catalytic pocket; and residues R307 and E313 of the D-loop, which are part of NBD1 and bring together the two NBDs upon binding of ATP (Fig. 4a–b; Supplementary Fig. 10). Pdr5 can utilise other nucleotides to fuel its transport activity[14,47], and this can be rationalised by our structures from the distribution of nucleotide contacts: the nucleobase does not appear to directly interact with amino-acid residues of the binding site (Supplementary Fig. 10).

**Pdr5 substrate rhodamine 6G binds in the cleft between the TMDs**. We solved the molecular structure of Pdr5 with the ligand R6G, a common organic dye. R6G is nestled between the two TMDs, in the inward-facing conformation of Pdr5, with ADP in the catalytic NBS and ATP in the deviant site (Fig. 4e). The amino-acid residues surrounding the ligand do not change positions considerably compared to the ADP-bound state (Supplementary Fig. 10c). Because of this, we hypothesise that there is no cross-talk between the substrate binding pocket and the rest of the transporter in the inward-facing conformation, which could be supported by the fact that Pdr5 is an uncoupled transporter[22].

R6G occupies a cleft that is positioned between the halves of the transmembrane portion of the transporter and is flanked by helices TMD1-TH2 and -TH5a as well as TMD2-TH8 and -TH11a (Fig. 4e cf. Fig. 1d). This includes residue S1360 which is known from mutational studies to be involved in R6G binding[48]. The pocket makes a large opening that reaches more than halfway down the depth of the cell membrane and is accessible from the cytosol as well as the inner leaflet of the lipid bilayer (Fig. 6a). The surface of the pocket is a mixed hydrophobic/hydrophilic environment, with the predominance of the former (Fig. 4e). The studied transport substrates of Pdr5 include mostly lipophilic molecules, which seem compatible with the pocket, although it has been shown that substrate volume rather than lipophilicity is the major determinant of transporter selectivity[7,49]. The entrance to the substrate cavity on the side of the bilayer has a slight negative charge, opposite to the positively charged R6G, whilst the cavity itself remains more neutral electrostatically (Fig. 4e). The location of R6G in the cleft between the TMDs is comparable to the positioning of substrates in other multi-drug transporters, such as ABCG2[50,51] (Supplementary Fig. 10d) or TmrAB[27]. This suggests that this pocket, a feature of the inward-facing conformation of Pdr5, is the transporter's substrate entry cavity (Supplementary Fig. 11).

**Vanadate-trapped Pdr5 adopts the outward-facing conformation**. The apo-Pdr5, ADP-Pdr5, and R6G-Pdr5 are in inward-facing conformations and are very similar (Supplementary Fig. 12a–b); the differences are limited to small structural rearrangements around the nucleotide-binding sites when ADP and ATP are present. Our model of AOV-Pdr5, a product of vanadate-trapping of Pdr5, yielded a structural model that is in the outward-facing conformation (Fig. 6, Supplementary Fig. 12b and Supplementary Movie 1). ADP and $V_i$, when bound in the canonical NBS1, and supported by the presence of ATP in the deviant NBS1, cause dimerisation of the two NBDs.

Pdr5 thus follows a pattern that is almost universally conserved amongst ABC transporters and other proteins that share a similar fold[1,11]. However, the closure of the NBS around the nucleotide is markedly asymmetrical in this transporter, occurring mainly through the interaction of the nucleotide bound in NBS1 with the D-loop of the NBD1. Between the inward- and outward-facing conformations, measured as the change in the distance from Walker A to the signature motif, NBS1 closes by 11 Å, while NBS1 just by 3.8 Å. As in other ABC transporters, the interaction and resultant dimerisation of the NBDs cause a structural change in both the transmembrane and extracellular domains, switching the transporter's overall conformation from inward-facing to outward-facing (Fig. 6).

In the outward-facing conformation, the helices of the TMD1 and TMD2 tilt towards each other on the cytosolic side and spread further apart on the extracellular side (Fig. 6b, Left). Helices that line the substrate entry cavity (TMD1-TH2, and TMD2-TH8, -TH11a) tighten around it, simultaneously opening up on the extracellular side to reveal a cleft that joins up with the newly expanded cavity between ECD1 and ECD2 (Fig. 6b, Right). The conformational state changes are also highly asymmetric. NBD2, which contains the catalytic nucleotide-binding site, performs a greater domain movement than NBD1, which hosts the deviant site—such asymmetry of movement has also been observed in the heterodimeric transporter TM287/288[52]. Similarly, greatest conformational changes in the transmembrane region of Pdr5 involve TMD2, which is responsible for the closure of the substrate cavity and the opening of the extracellular part of the transporter, while TMD1 is comparatively invariant (Supplementary Fig. 12b). The single-site hydrolysis is a simplification of the symmetric ABC transporter system, in which two ATP molecules (one in each catalytic site) are required to complete the transport cycle[53].

The use of vanadate to mimic the transition state of nucleotide hydrolysis allows the transporter to be captured in a state that would otherwise be inaccessible due to the speed of hydrolysis[24–27]. Various studies report a high degree of structural similarity between the transition and the pre-hydrolytic states for other ABC transporters, indicating that ATP binding and not hydrolysis induce the conformational change in the transporter from the inward-facing to the outward-facing state[27,54–58]. We suspect that a similar effect is at play in Pdr5, but further work would be needed to establish this. Also, the details of the substrate passage through the channel between the TMDs remain to be elucidated for Pdr5 (and indeed for other ABC transporters), but we envisage it to be somewhat akin to a peristaltic action where the closure of the entry channel on the cytoplasmic side pushes the substrate towards the expanding exit channel (Supplementary Movie 2). A similar mechanism has been suggested for other members of the ABCG subfamily, such as ABCG2[39].

The access to both the entrance cavity and the exit channel is from the membrane-cytosol and membrane-extracellular interface (Fig. 6 and Supplementary Fig. 11), which in principle can accommodate substrates coming from and/or directed to a lipophilic or hydrophobic environment; this appears compatible with the amphipathic character of reported Pdr5 substrates and implies that substrates are not released into the extracellular space but rather into the outer leaflet of the plasma membrane. The entrance and exit sites are asymmetric in shape and surface properties, so that the probability of entry from one site may be higher than from the other (Fig. 6). The Pdr5 exit channel might also contain the equivalent of the leucine gate found in ABCG2, which controls drug extrusion in that protein[59] (Supplementary Fig. 13).

**ATP binding induces rearrangement in the N-terminal extension of Pdr5**. The N-terminal part of Pdr5 contains a

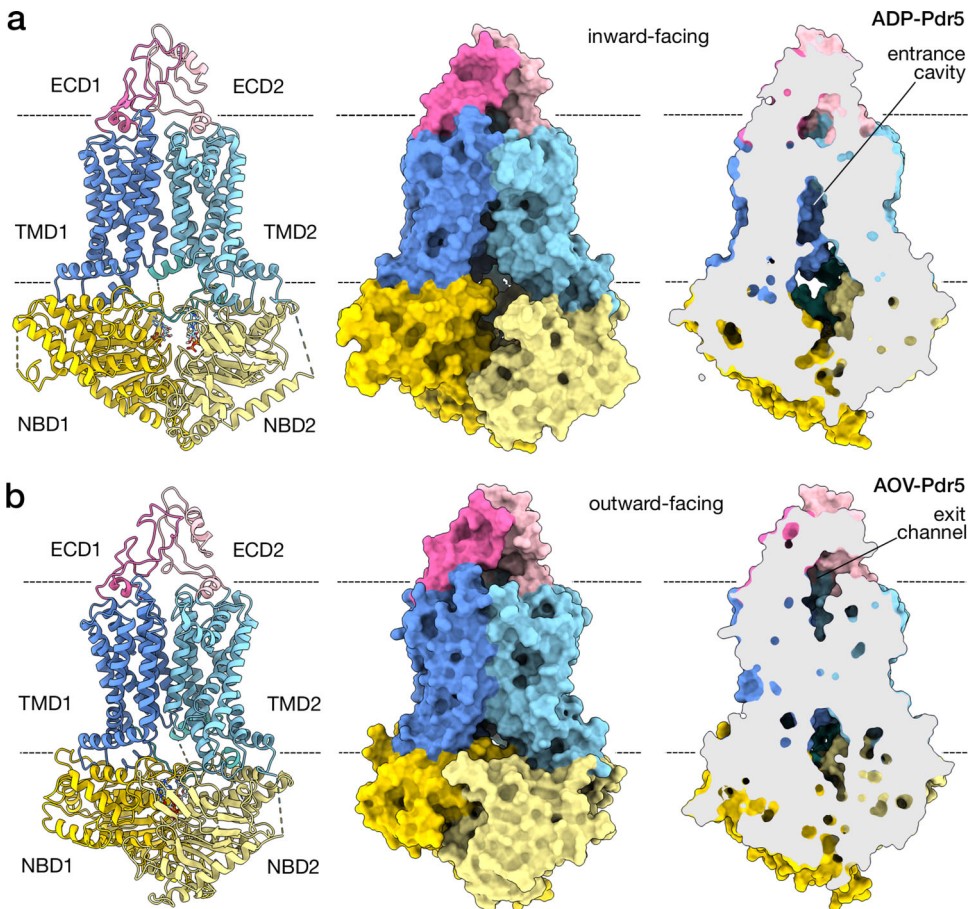

**Fig. 6 Conformational changes upon ATP binding.** The two panels depict the molecular structure model of inward-facing (**a**) and outward-facing (**b**) conformations of Pdr5 in cartoon representation, and as a solvent-excluded surface with its cross-section. **a** Shown is the Pdr5 in the post-hydrolytic state, with ADP present in the catalytic site (ADP-Pdr5) and the transport channel open onto the cytoplasmic side. The position of the substrate entrance cavity is indicated. **b** The Pdr5 in the vanadate-trapped, outward-facing state is depicted. The exit channel for the substrate, which opens onto the extracellular side, is indicated as well. Abbreviations: ECD, extracellular domain; NBD, nucleotide-binding domain; TMD, transmembrane domain.

stretch of some 120 amino-acid residues that precede the conserved portion of the NBD domain and is characteristic of the PDR subfamily. Mutagenesis data imply a role of this domain in signalling to other cellular processes and/or trafficking. We observe that the N-terminal extension contains two helices H1 and H2 connected by an unresolved region of 43 amino-acids (Supplementary Fig. 5 and Supplementary Fig. 6). The two helices are on the outside of NBD1 (top left of Fig. 1c; cf. Supplementary Fig. 12d) and occupy well-defined positions in the inward-facing state (ADP-Pdr5). In contrast to the rest of the NBD1, which maintains its overall structure between the two states, a pronounced structural change in the N-terminal extension is observed with ATP binding and switch to the outward-facing (AOV-Pdr5) conformation (Supplementary Fig. 12d). In the outward-facing state, helix H1 appears to be disordered and is displaced by a coupled movement of H2 and helix H15 (Supplementary Fig. 12d). These conformational changes could support recognition to trigger downstream effector processes.

**Drug efflux occurs as Pdr5 switches between two conformations.** The structural models of Pdr5 in different nucleotide and ligand states presented here suggest how efflux works (summarised graphically in Fig. 7). In the resting state, Pdr5 adopts an inward-facing conformation with the substrate entry channel open towards the cytoplasm and inner leaflet of the membrane. In this state,

corresponding to our ADP-Pdr5 model, the transporter has an ADP molecule in NBS2 and ATP in the deviant NBS1. On the basis of cellular concentrations of ATP in *S. cerevisiae* ($1.51 \pm 0.32$ mM[60]) and the transporter's $K_M$ for the nucleotide (1.7 to 1.9 mM in vivo[12,44], $0.44 \pm 0.05$ mM in vitro[14]), it seems likely that the apo state may represent about 50% of the population. The other Pdr5 states, which were prepared from samples with added ATP, have the nucleotide bound in the non-catalytic, deviant binding site; this would support earlier suggestions that ATP remains in the site throughout the transport cycle[34] under physiological conditions.

The substrate entry cavity of the inward-facing Pdr5 can receive the efflux substrate, as evidenced by our structure of Pdr5 with R6G (R6G-Pdr5). At this point, if ADP in the catalytic site is exchanged for ATP, transport can occur. The binding of ATP induces a structural change, which involves the transporter switching conformation from inward-facing to outward-facing, as exemplified by our vanadate-trapped structure (AOV-Pdr5). This change involves sealing off the substrate entry cavity and opening an exit channel on the extracellular side of the transmembrane domain. The transport substrate moves from the entrance cavity into the exit channel and is released into the extracellular medium or outer leaflet. After the nucleotide is hydrolysed, the transporter reverts to the inward-facing conformation. In this state, ADP remains bound to the catalytic site, and the substrate entry channel is re-opened. Thus, the transport cycle of Pdr5 is completed, and the transporter can accept the next molecule of the substrate.

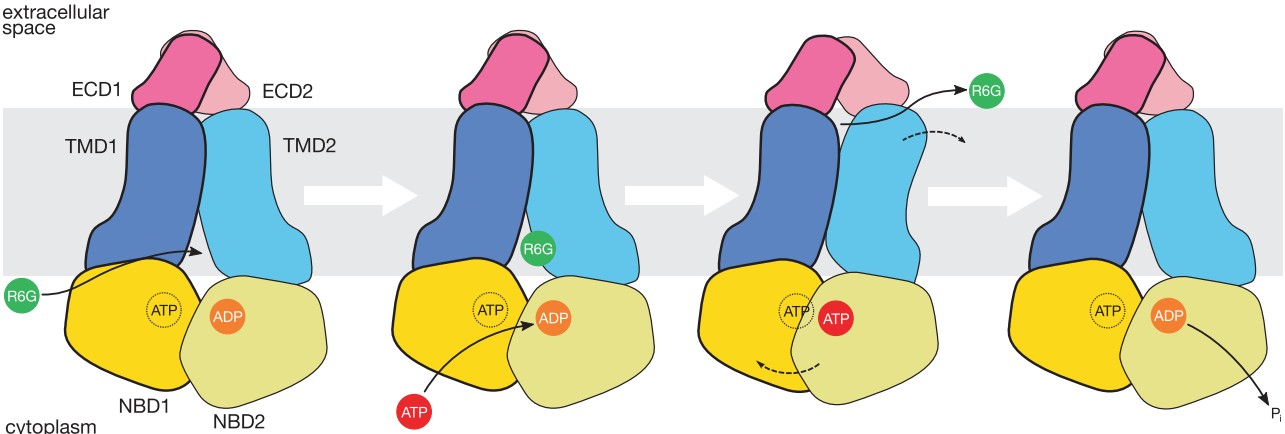

**Fig. 7 Transport cycle of Pdr5.** The substrate (in this case, rhodamine 6G) enters the cavity between the transmembrane domains from the cytoplasmic side when Pdr5 is in inward-facing conformation. ADP remaining in the catalytic site from the previous hydrolysis step is exchanged for ATP. ATP binding triggers a conformation change in Pdr5 from inward- to outward-facing, whereby the substrate is pushed through the substrate channel and released into the extracellular medium. Upon nucleotide hydrolysis, the transporter reverts to inward-facing conformation, ready to receive another molecule of substrate. In cellular conditions, an ATP molecule remains bound to the inactive site throughout the cycle. The asymmetric nature of nucleotide hydrolysis means that one half of Pdr5 makes a larger conformational move than the other. Abbreviations: ECD, extracellular domain; NBD, nucleotide-binding domain; R6G, rhodamine 6G; TMD, transmembrane domain.

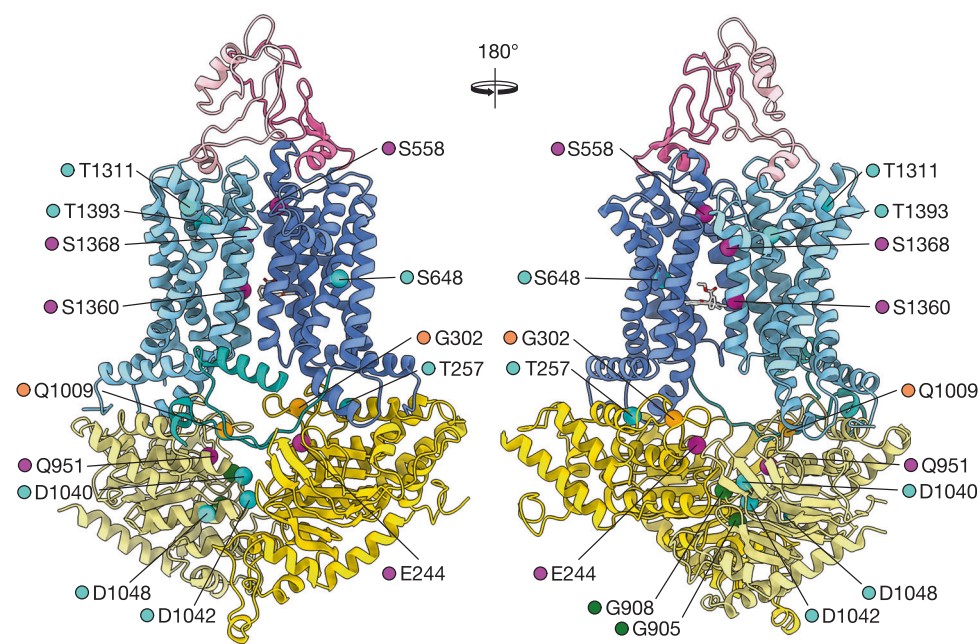

**Fig. 8 Mapping of known mutations of Pdr5.** Functional mutants of Pdr5 found in literature (see: Supplementary Table 5) are mapped onto the cartoon representation of inward-facing Pdr5 structure (R6G-Pdr5). Front and back views are shown. Mutations are highlighted as coloured spheres centred on the $C_\alpha$ of the residue, while the bound ligand, rhodamine 6G, is shown in sticks representation. Mutations that inactivate Pdr5 are labelled with green spheres, those that confer or alter drug resistance with orange and cyan spheres, respectively. Those with other, different phenotypes (see: Supplementary Table 5) are highlighted with magenta. Colour-coding of domains of Pdr5 is identical to Fig. 1 and used throughout.

**Pdr5 structure can rationalise many observed mutational phenotypes.** Our structures can also provide the basis to explain important functional mutants that can be found in the rich Pdr5 literature. We analysed a number of these mutants (Supplementary Table 5) and mapped them onto the inward-facing structure of Pdr5 (Fig. 8). Mutations that inactivate the transporter, G905S and G908S, are located in the Walker A motif of the active NBS1. An exchange of the highly conserved Gly residues in P-loop NTPases is a well-known example for the inactivation of these enzymes, for example in Ras[61]. The other known mutations within the NBD are also localised in conserved motifs, such as the Q-loops of NBD1 (E244Q) and NBD2 (Q951G), the X-loops

of NBD1 (G302D) and NBD2 (G1009C) and the D-loop of NBD2 (G1040D and D1042N). It is obvious that these mutations will interfere with ATP binding/hydrolysis and/or NBD–NBD communication, leading to changed phenotypes. Interestingly, G1009 interacts with I805 of the MKEGEIL motif via the backbone. A mutation to Cys would thus abolish this interaction and interfere with domain-domain communication in Pdr5 resulting in generally reduced drug resistance[23].

Mutations within the TMD can be grouped into two categories. The first category involves mutations of residues that are part of the R6G binding site (S1360F) or its vicinity (S1368A and T1391I). The other category comprises residues that have a more

structural function: S558Y of TH2 for example interacts with and positions the linker between TH11a and TH11b. S648 of TH4 is located in a helical bundle with TH5c and TH6. The bulky Phe residue in the S648F mutation likely disturbs the bundle architecture. Similarly, Y1311 of TH9b is located in a cluster of Tyr residues (1301, 1305, 106, and 1311) and a replacement with Ser will destabilise the aromatic interaction, resulting in an altered drug specificity. No structural explanation, however, can be provided for the phenotypes of T257I and S1048V.

Besides providing context to past functional and biochemical studies on Pdr5, this work presents the first molecular structure of an ABC transporter from the PDR subfamily. Our models illustrate the unique features of this group of proteins. Chief amongst them is the dynamic linker domain. We were able to assign a functional role to the linker and show how it can complement the binding of ATP in the non-catalytic, degenerate NBS—another characteristic of PDR transporters. We also show how asymmetry in nucleotide hydrolysis is reflected in the conformational cycle of the transporter. The first half of Pdr5 (NBD1-TMD1) stays relatively immobile compared to the second, which contains the active, catalytic nucleotide site. Between the two halves of the transporter lies the transport channel, able to accommodate the wide chemical variety of Pdr5 substrates. The asymmetry of PDR is reflected in its structure again: the substrate channel contains a single exit site. However, this paper provides more than just a static glimpse into the Pdr5 function. Through our study of multiple conformational states, we were able to propose a model of how this protein actively pumps chemicals across the membrane, using ATP hydrolysis. Pdr5 is an archetypal member of a unique subfamily of proteins, but it is also, unmistakably, an ABC transporter, in both function and structure.

## Methods

**Protein expression and purification.** The protocol of Wagner et al.[14] was used for producing and purifying stable Pdr5 preparations from *S. cerevisiae* and is summarised briefly here.

Histidine-tagged Pdr5 was purified from *S. cerevisiae* strain YRE1001[22] grown in YPD medium at 30 °C. Cells were harvested by centrifugation at $OD_{600} = 3.5$ and lysed with glass beads in buffer containing 50 mM Tris-HCl (pH = 8.0), 5 mM EDTA and EDTA-free Protease Inhibitor Cocktail tablets (Roche AG, Basel, Switzerland). Cell debris was removed by centrifugation (4 °C): twice at $1000 \times g$ for 5 min and once at $3000 \times g$. Cell membrane was pelleted from the resulting supernatant by centrifugation at $20,000 \times g$ for 40 min (4 °C). The cell membrane pellet was resuspended in Buffer A: 50 mM Tris-HCl (pH = 7.8), 50 mM NaCl, 10% (w/v) glycerol and adjusted to 10 mg mL⁻¹ total protein concentration. The solubilisation of membrane proteins was achieved through the addition of 1% (w/v) of *trans*-4-(*trans*-4′-alkylcyclohexyl)cyclohexyl-α-D-maltoside (*trans*-PCC-α-M; Glycon Biochemicals, GmbH, Luckenwalde, Germany) and 1 h incubation at 4 °C, with gentle stirring. Non-solubilised material was removed by centrifugation at $170,000 \times g$ for 45 min (4 °C).

Pdr5 was separated from other proteins by immobilised metal ion affinity chromatography (IMAC), using a HiTrap Chelating HP column (GE Healthcare, Chicago, IL, USA) loaded with $Zn^{2+}$ ions and equilibrated with low-histidine buffer: 50 mM Tris-HCl (pH = 7.8), 500 mM NaCl, 10% (w/v) glycerol, 2.5 mM L-histidine, 0.003% w/v *trans*-PCC-α-M. The sample was loaded onto the column, washed, and eluted with a concentration gradient of histidine (up to 100 mM). Fractions of IMAC eluate containing Pdr5 were pooled and concentrated using a Vivaspin 6 centrifugal concentrator unit (Sartorius AG, Göttingen, Germany) with a molecular weight cut-off (MWCO) of 100 kDa. Concentrated protein was further purified using size-exclusion chromatography (SEC) on a Superdex 200 10/300 GL (GE Healthcare, Chicago, IL, USA) column containing 0.003% (w/v) *trans*-PCC-α-M.

Fractions of highest Pdr5 concentration (approx. 2.5 mg mL⁻¹) from SEC were selected for reconstitution into peptidiscs[17]. The peptidisc solution was prepared from bulk lyophilised peptidisc (Peptidisc Biotech, Vancouver, BC, Canada) dissolved in 20 mM Tris-HCl (pH = 8.0) to a final concentration of 10 mg mL⁻¹. Dissolved peptidisc was mixed with solubilised Pdr5 sample in 1:1 weight ratio (1:38 molar ratio) and incubated for 30 min at room temperature. The mixture was separated using SEC on a Superose 6 10/300 GL column (GE Healthcare, Chicago, IL, USA) equilibrated with buffer containing 50 mM Tris-HCl (pH = 7.8), 100 mM NaCl and no detergent. This step yielded homogenous Pdr5/peptidisc assemblies.

**Liquid drug assay.** Ketoconazole, fluconazole (Sigma-Aldrich Merck KGaA, Darmstadt, Germany), rhodamine 6 G (Acros Organics, Fair Lawn, NJ, USA) and cycloheximide (Honeywell Fluka Inc., Charlotte, NC, USA) were prepared as stock solutions in dimethyl sulfoxide (DMSO, Acros Organics, Fair Lawn, NJ, USA) and diluted in sterile water, as necessary. The assay was carried out in sterile 96-well microtiter plates with 20 µL drug, 180 µL YPD medium and 50 µL of $OD_{600}$ 0.2 yeast culture. Plates were incubated at 30 °C for 48 h and $OD_{600}$ was measured with a microplate absorbance reader (Bio-Rad Laboratories Inc., Hercules, CA, USA).

**ATPase measurements in Pdr5-enriched plasma membranes.** Plasma membranes were prepared, and ATPase activity was determined as described previously[22]. Briefly, oligomycin (OM) sensitive ATPase activity of plasma membrane fractions was measured by a colorimetric assay. Isolated plasma membranes (0.1 or 0.2 µg per well) were incubated with 0.1–10 mM ATP, 5 mM MgCl₂ in 300 mM Tris-glycine buffer (pH = 9.0) in a final volume of 100 µL. To reduce background, 0.2 mM ammonium molybdate, 50 mM KNO₃, and 10 mM NaN₃ were added. In a control reaction, OM (20 µg mL⁻¹) was added to the assay to determine the OM-sensitive ATPase activity. After incubation at 30 °C for 20 min, the reaction was stopped by adding 25 µL of the sample to 175 µL of ice-cold 40 mM H₂SO₄. The amount of released inorganic phosphate was determined by a colorimetric assay, using Na₂HPO₄ as standard.

**Statistical analysis.** All experiments were performed in triplicates. Error bars in figures represent mean ± standard error. Ordinary one-way ANOVA tests were performed as part of the analysis. *P*-values are provided in the legends of the corresponding figures and tables, respectively. Degrees of freedom were 18 (Supplementary Fig. 7 and Supplementary Table 3) and between 29 and 34 (Supplementary Fig. 8B and Supplementary Table 4). *F* values against the four drugs analysed were 20.7 (cycloheximide), 37.1 (fluconazole), 9.4 (ketoconazole), and 3.4 (rhodamine 6G). In the case of the mutants of the linker domain, *F* value were calculated to be 3.1 for $K_M$ and 19.8 for $V_{max}$.

**Cryo-EM sample preparation and data collection.** To prepare the vanadate-trapped state sample (AOV-Pdr5), Pdr5 resuspended in peptidiscs (1.5 mg mL⁻¹) was mixed with 2 mM of ATP and 2 mM MgCl₂, and incubated for 2 min. To this solution, 200 µM of freshly prepared sodium orthovanadate (Na₃VO₄) was added and incubated for further 2 min. In the case of the R6G-Pdr5 dataset, Pdr5 was mixed with 2 mM of ATP, 2 mM MgCl₂, and 100 µM of rhodamine 6 G solution in water. In the case of the ADP-Pdr5, just 2 mM of ATP, 2 mM MgCl₂ were added. The samples were incubated for 20 min prior to grid freezing. No additives were used to prepare the apo state sample. To improve the distribution of particles in vitreous ice (Supplementary Fig. 3c), an aqueous solution of CHAPSO was added to all the samples to final concentration 8 mM. The solution was mixed by pipetting and used immediately in grid preparation. 3 µL of the sample mixture were applied onto a Cu 300-mesh EM grid with R1.2/1.3 holey carbon support film (Quantifoil Micro Tools GmbH, Großlöbichau, Germany) which had been glow-discharged prior to use. The sample applied to the grid was vitrified in liquid ethane at –180 °C using a Vitrobot Mark IV (Thermo Fischer Scientific Corp., Eindhoven, The Netherlands).

The grids were imaged using Titan Krios G3 TEM (Thermo Fischer Scientific Corp., Eindhoven, The Netherlands) operated at liquid nitrogen temperature, 300-kV accelerating voltage, and in EF-TEM mode with GIF slit size of 20 µm. The micrographs were recorded using automated image acquisition software EPU (Thermo Fischer Scientific Corp., Eindhoven, The Netherlands), in the case of the ADP-Pdr5, R6G-Pdr5, and AOV-Pdr5 samples, on a K3 direct electron detector (Gatan Inc., Pleasanton, CA, USA) in super-resolution counting mode, at a detector pixel size of 0.326 Å (nominal magnification ×130,000) and nominal defocus values between –0.7 and –2.2 µm. The micrographs of the apo-Pdr5 sample were collected with a K2 Summit direct electron detector (Gatan Inc., Pleasanton, CA, USA) in counting mode, at a detector pixel size of 1.07 Å (nominal magnification ×130,000) at similar nominal defocus values. For the apo-Pdr5 sample, exposures lasted 12 s and were collected in 40 fractions (processing frames) each, with total measured electron fluency of exposure was 58.1 e⁻ Å⁻². A total of 4,211 micrographs were acquired over the course of a single 72-h data collection session. For the ADP-Pdr5 sample, the exposures lasted 1.31 s, collected in 49 fractions and fluency of 47.21 e⁻ Å⁻². This dataset contained 3,415 micrographs. The R6G-Pdr5 and AOV-Pdr5 datasets were obtained using similar exposure conditions, with fluencies of 47.73 and 48.13 e⁻ Å⁻²; they amounted to 3604 and 8370 micrographs, respectively.

**Cryo-EM data processing for structure solution.** The cryo-EM map of apo-Pdr5 was reconstructed in RELION-3.0[28] and cryoSPARC-2.15[62] (Supplementary Fig. 4). Micrographs were corrected for beam-induced sample motion within RELION. Contrast transfer function (CTF) parameters were estimated using Gctf-1.08[63] on non-dose-weighted micrograph averages. A subset of the images with estimated maximum resolution below 4 Å was selected from the full dataset for further processing. A manual pick of ca. 2000 particles was used to prepare a set of 2D references for automated particle search, which was subsequently optimised to

yield diverse projections of Pdr5 particles. The initial particle pool was extracted in 320-pixel boxes and pruned through a series of reference-free 2D classification rounds to 68,154. A model for cryo-EM reconstruction was created through stochastic gradient descent (SGD) algorithm in RELION on 8,000 randomly selected particles. The model and the pruned particle pool were subjected to 3D auto-refinement, which generated the first intermediate cryo-EM map used for subsequent Bayesian particle polishing and CTF refinement[64].

To improve the quality of the protein part of the apo-Pdr5/peptidisc assembly cryo-EM map, the density of the peptidisc was subtracted from the full map, using tools and procedures implemented in RELION[65,66]. First, a mask template was prepared by removing the Pdr5 part of the density from the full map, with the Volume Eraser utility of UCSF Chimera-1.13[67]. Second, the template was transformed into a binarised mask with a cosine-soft edge in RELION. The signal from within the mask was subtracted from the particles in the full map set from the last 3D auto-refinement step. The result of the subtraction was validated with 2D classification of the subtracted particle set. The final map reconstruction was performed on subtracted particles using non-uniform refinement (NU-refinement) algorithm in cryoSPARC[68]. The overall resolution of the obtained map, as judged by Fourier-shell correlation (FSC) of the half-maps with the gold-standard $FSC_{0.143}$ criterion[29], was 3.45 Å.

The cryo-EM maps from the ADP-Pdr5 and the rhodamine 6G R6G-Pdr5 datasets were processed similarly, although RELION-3.1 was used for the reconstruction and corrected for high-order aberrations and anisotropic magnification errors[69]. The raw micrographs were down-sampled by a factor of 2, yielding an input set of images with an effective pixel size of 0.652 Å. Contrast transfer function (CTF) parameters were estimated using CTFFIND-4.1[70] on non-dose-weighted micrograph averages. The initial particle pool was extracted in 524-pixel boxes and pruned through a series of 2D and 3D classification rounds. In the case of the ADP-Pdr5 dataset, an improvement in the signal-to-noise ratio of the protein part of the Pdr5/peptidisc assembly was achieved using SIDESPLITTER-1.2[71] which locally de-noises cryo-EM maps[72] and 3D map reconstructions were done externally to RELION-3.1 between the iterative steps of 3D auto-refinement. The overall resolution of the final maps obtained by the combined use of SIDESPLITTER-1.2 and RELION-3.1 was 2.85 Å (ADP-Pdr5) and 3.13 Å (R6G-Pdr5), per the $FSC_{0.143}$ criterion.

The vanadate-trapped AOV-Pdr5 dataset was processed similarly to apo-Pdr5, combining the use of RELION-3.1 and cryoSPARC-3.0. The raw micrographs were down-sampled to an effective pixel size 0.652 Å. The initial particle pool was extracted in 524-pixel boxes and pruned through a series of 2D and 3D classification rounds. The overall resolution of the final map was 3.75 Å, per the $FSC_{0.143}$ criterion.

**Model building and refinement**. The reconstructed cryo-EM map of ADP-Pdr5 was used for de novo building of an atomic model of *S. cerevisiae* Pdr5. A preliminary model was calculated using the Rosetta software suite[73]. The model was adjusted manually in Coot-0.9[74], part of the CCP-EM package[75], and in ISOLDE-1.0b3[76]. Model refinement was carried out in real space using PHENIX-1.18.2[77]. To assist model building, the maps were subjected to a density modification procedure[78], implemented as part of the PHENIX-1.18.2. Model building in high flexibility regions of Pdr5 was facilitated through local map sharpening with LocScale[79], part of the CCP-EM-1.4.1 program suite[75]. Other models were built manually, using the ADP-Pdr5 structure as a starting point. Locally sharpened maps were also used for visualisation purposes. Molecular graphics were rendered in ChimeraX-1.13[80] and PyMOL-1.8.2.3 (Schrödinger LLC). All refinement and data collection statistics are summarised in Supplementary Table 1.

**Reporting summary**. Further information on research design is available in the Nature Research Reporting Summary linked to this article.

## Data availability
The molecular structure data generated in this study have been deposited in the PDB database under the accession codes (PDB-IDs): 7P03, 7P04, 7P05, 7P06. The cryo-EM map data are available via EMDB: EMD-13142, EMD-13143, EMD-13144, EMD-13145. Structural comparisons presented in this paper utilised publicly available data from the PDB: 5DO7, 6XVI, 7K2T, 6JBH, 6HCO. Source data are provided with this paper.

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

## Acknowledgements

A.H., D.D. and B.F.L. are supported by European Research Council (ERC, VisTrans, grant no. 742210). The Center for Structural Studies, HHU Düsseldorf, is funded by Deutsche Forschungsgemeinschaft (DFG, grant no. 417919780). We thank the staff of the BiocEM facility at the Department of Biochemistry, University of Cambridge, Dr. Dimitri Y. Chirgadze, Dr. Steven Hardwick, and Mr. Lee Cooper for assistance with cryo-EM data collection.

## Author contributions

B.F.L. and L.S. designed the study; A.H., M.W., D.D., S.R., L.-M.N. and B.F.L. performed biochemical experiments; A.H., S.H.J.S. and H.G. supervised M.W., L.-M.N. and S.R.; A.H., D.D. and B.F.L. collected cryo-EM data; A.H. and B.F.L. processed and analysed the cryo-EM data; A.H., B.F.L. and L.S. wrote the manuscript, and all authors revised it.

## Funding

## Competing interests
The authors declare no competing interests.
