## [Peer Review File · Nature Communications]

REVIEWER COMMENTS

Reviewer #1 (Remarks to the Author):

In this manuscript, Harris et al. present four cryo-EM structures of peptidisc-reconstituted Pdr5, *S. cerevisiae* ABC transporter, belonging to the pleiotropic drug resistance family. The authors provide a detailed description of those structures, emphasizing structural features, important for the transporter function. The structures, described here, are a valuable addition to the ABC transporter gallery, and the main findings, including asymmetric conformational changes and the substrate-binding mode, are of high interest to the wide scientific community. Cryo-EM analysis meets the current standards, however, there are many flaws in the representation and description of the data, which need to be fixed:

1. The summary lacks details – it summarizes the main findings, but does not even tell which method was used to get them. The authors' statement "structural and functional results reveal details of an ATP-driven conformational cycle" leaves me wondering what kind of functional results the authors are referring to. Actually, there is not a single experimental evidence for the biochemistry in this paper, which is largely unexpected. Authors should at least show ATPase activity assays to demonstrate that the transporter is active after reconstitution into peptidiscs. Information on the hydrolysis speed would additionally support the findings that Pdr5 hydrolyzed most of ATP prior to plunge-freezing of EM grids. Moreover, the statement in the summary about the coupling of nucleotide hydrolysis with drug efflux is not correct as Pdr5 is an uncoupled transporter, which authors also mention themselves in the text later.
2. For cryo-EM analysis, Pdr5 was reconstituted in peptidiscs, but the authors do not include any figure showing this. At least an SDS-PAGE gel, proving that reconstitution was successful, should be shown. In general, it is also recommended to include a SEC profile into supplementary figures to provide insights into sample homogeneity.
3. On the second paragraph of page 4, the authors write: "The resolution of the maps ranged between 2.8 Å and 3.5 Å", but according to Table S2, the resolution of maps is between 2.9 and 3.8 Å. This should be fixed accordingly, and the deposition codes for the maps and models should be included into the manuscript.
4. As this is an entirely structure-based manuscript, it is necessary to provide maps and models for evaluation by the reviewers.
5. The authors do not see magnesium at the non-catalytic site in their structures. While this is very likely for the inward-facing structures, how sure are the authors about the absence of magnesium in the AOV-Pdr5 structure, which has the lowest resolution? I have some doubts about making such claims at 3.8 Å resolution. The authors should support these claims in the supplementary figures by demonstrating model and map overlay at the nucleotide-binding sites for all the structures.
6. There are still debates in the field about conformational changes following ATP binding and hydrolysis, therefore, simply dropping a statement like in the subtitle on page 8 ("ATP binding induces a change to outward-facing conformation of Pdr5") without a solid proof is not a good idea. I am sure that this statement is correct, but it should not be used here as the data does not show this - the authors do not have ATP bound outward-facing structure; they only have ADP-vanadate structure, which is not the same.
7. In the methods, the authors should write at which concentration Pdr5 was used for cryo-EM grid preparation.
8. The authors mention that they used CHAPSO to improve particle distribution, however, they do not show the angular distribution plots. Such plots should be added to the supplementary data. I would also recommend adding the processing workflow to the supplementary figures.
9. The representation of the structures is very confusing. It is generally accepted to display cytoplasmic side on the bottom and the extracellular side on the top, but the authors display everything inverted. I am not sure if the authors wanted to show their creativity or to highlight the reverse transmembrane topology, but such representation simply brings confusion and, therefore, should be changed to promote direct comparability with other studies. There is enough confusion in the field regarding nomenclature, naming of the different states and other things, so it is really a bad

idea to introduce yet another way of representation. A good figure should deliver the main message without the need of additional text, which is absolutely not the case in the figures of this manuscript. But I do like that the authors display the peptidisc.

10. The authors should be consistent with their naming – in panels E-F of Figure 1, the ADP-Pdr5 map is called ATP-Pdr5. At the end of the caption of the same figure, the authors write “Arrowheads point to selected amino acid residues”, but it is not clear why these amino acid residues were selected. In general, the authors should be more specific in figure captions, for example, in Fig. S1 it is written: “Figure panels illustrate some of the steps in the structural reconstruction of apo-Pdr5 (A–D), ADP-Pdr5 (E–F), AOV-Pdr5 (G–H), and R6G-Pdr5 (I–J).” This is too unspecific and also implies that the authors display the same steps for all four structures, but this is not the case here. I would suggest removing this sentence and simplifying the caption by explaining which step and for which protein is shown on each panel.

11. Fig. 2 lacks consistency – it shows NBS1 and NBS2 for ADP-Pdr5, but only NBS2 for AOV-Pdr5 structure. The authors should also include NBS1 for AOV-Pdr5 here. It would be a good idea to show the densities for nucleotides and the substrate in this figure, but if it gets overcrowded, such densities with corresponding fits should be at least shown in supplementary figures as requested before (see comment 5).

Reviewer #2 (Remarks to the Author):

As a mammalian p-glycoprotein homolog in *Saccharomyces cerevisiae*, Pdr5 has attracted a lot of attention. This work elucidates the molecular structure of Pdr5 for the first time based on Cryo-EM and addresses a key problem in Pdr5 research. It will be very helpful for the further studies on structural and functional relation of Pdr5.

Reviewer #3 (Remarks to the Author):

Harris et al. present here a nicely conducted structural study on the well-characterized and “famous” yeast ABC transporter Pdr5. Four cryo-EM structures are presented, three (apo, ADP-bound and ADDP+R6G-bound) in inward-facing states and one (vanadate trapped) in the outward-facing state. Together with the structural data on drug binding (Rhodamine-6G), this structure series allows to formulate a transport mechanism (which in fact agrees with transport mechanisms of type II ABC exporters postulated earlier).

While these are not the first structures of a type II ABC exporter, the work nevertheless provides highly important novel insights into structural peculiarities of Pdr5 (and other members of the pleiotropic drug resistance transporter subfamily), namely a nucleotide-sensing linker domain, a unique extracellular domain, highly asymmetric nucleotide binding sites and finally an extra N-terminal domain with unknown function.

The structural work appears mostly solid (see some mild criticism below) and the manuscript is well written and intelligible to readers outside of the ABC transporter community. The figures are informative, aesthetical and clear. Provided the authors generate at least some extra functional data (see suggestions below) and place the work better into the larger context of ABC transporters (again suggestions are listed below), I consider this manuscript as a very good match for Nature Communications.

Major points

1) Given the structural similarities to ABCG2, Pdr5 (at least in this reviewer’s view) belongs as well to the same ABC transporter subclass, namely it is a type II ABC exporter (older nomenclature) or a type V ABC transporter (newer nomenclature). The manuscript in this form falls short in providing any information regarding the structural classification of Pdr5, which very likely will lead to confusion in the field (currently, the message comes across that Pdr5 is the founding member of a new ABC

transporter subclass not structurally investigated thus far).

2) Assuming that Pdr5 belongs to the type II ABC exporters, it is very important to highlight similarities and differences between the existing and these novel structures. A comprehensive list of structures can be found here <https://febs.onlinelibrary.wiley.com/doi/10.1002/1873-3468.13935>. It is worth noting that for ABCG2, both inward- and outward-facing structures had been determined, allowing for a very close examination of similarities and peculiarities. Further, it is worth to be noted that with the structure of ABCG5/8, a heterodimeric member of this class of transporters had been solved already once. A comparison to this ABCG5/8 structure (PDB: 5DO7) is certainly needed. Another interesting aspect is whether Pdr5 is related more closely to the eukaryotic members of the subfamily (ABCA1, ABCG2, ABCG5/8) or to the prokaryotic members of the subfamily (Wzm-WztN and TarGH).

3) The most exciting structural finding of this study is (in this reviewer's view) the nucleotide-sensing linker domain. The authors pinpoint a highly conserved motif (MQKGEIL) being critically involved in mediating important contacts. The authors should confirm with functional and biochemical experiments that this motif really plays a role and it has to be investigated, whether and how this motif is involved in nucleotide sensing at the molecular level.

4) The authors claim that they observed R6G binding. However, they do not provide cryo-EM densities to further support this claim (in fact, a comparison of the structures determined with ADP versus ADP+R6G are needed to judge). As for point 3, the authors should generate a few mutants in the R6G binding region and characterize them at the functional/biochemical level to further substantiate their findings (as a positive control, the authors can carry along a previously described mutant, namely S1360F).

5) Line 366/367: "These structures now also provide the basis to explain important functional mutants of Pdr5 (for a selection see Table S1)." These previously described mutants have to be discussed more thoroughly in a separate section.

6) The manuscript essentially lacks a conclusion (other than explaining the transport mechanism). In this reviewer's view, a conclusion could contain the novelty of findings of Pdr5 in the context of existing knowledge about the type II ABC exporter structures and mechanisms.

7) It may sound a bit silly, but why do the authors depict all structures "upside down"? It is common sense in the ABC transporter field to show the structures with the NBDs down and not up. Did this happen by "accident" or is it the intention of the authors to make Pdr5 look like very special?

Minor points

1) There are multiple errors in Table S1:

- "Q951Q"? ♦ Q951G
- Maybe explain a bit more why these different mutations modulate the function of Pdr5
- E244G was not found in publication 74
- S1360F was found in publication 76, but not in 77 in which they describe S1368A
- S1386A was not found in publication 78. In this publication they describe V656L, P596L, L1367F, and A670S.
- Please check again all citations

2) Is the transporter still active when reconstituted into peptidiscs? Of course, we see the different states, so in theory it must be active. However, it is of interest whether the transporter is less/similar/more active compared to in vivo and other in vitro studies? The in vitro study (14) the authors are referring to was performed with detergent-purified Pdr5.

3) Line 268-270: "Because of this, we surmise that there is no cross-talk between the substrate binding pocket and the rest of the transporter in the inward-facing conformation, which is supported by the fact that Pdr5 is an uncoupled transporter." However, there is ADP in the consensus ATP binding site when R6G is bound. From this, can one really assume that the binding pocket is not

communicating with the rest of the transporter? In the referred publication (22), drug-mediated inhibition of Pdr5-specific ATPase activity was observed, based on which it was claimed the transporter is uncloupled.

4) Line 233: is there a structural explanation, why magnesium was not found to be bound in the degenerate site?

5) Line 245: "The ADP-Pdr5 structure was solved using Pdr5 samples with added ATP, indicating that the majority of ATP was hydrolysed before flash-freezing in liquid ethane." Is this really plausible? How much ATP was added and how much Pdr5 was present? And what is the ATPase activity of Pdr5 in peptidisc at this temperature of incubation (see also point above). And how long was the transporter incubated before freezing? Based on these parameters, the authors can estimate how much ATP has been converted to ADP. We could imagine, that ATP turnover was only partial, but that ADP/ATP exchange at the canonical site might be slow?

6) Line 258: "Pdr5 can utilise other nucleotides to fuel its transport activity^{14,43}, and this can be rationalised by our structures from the distribution of nucleotide contacts." What exactly is meant with this? Which residues are likely of relevance for binding e.g. GTP versus ATP?

7) Line 296: "This occurs mainly through the interaction of the nucleotide, bound in NBS2, with the D-loop of the NBD1, and follows a pattern that is almost universally conserved amongst ABC transporters and other proteins that share a similar fold^{1,11}." Nucleotide sandwiching most likely has also occurred at the degenerate NBS1. At least in TmrAB, Mrp1 and TM287/288, both NBSs are completely closed. The way the authors describe the situation here, it implies that only the ATP at the consensus NBS2 glues the NBDs together. Fig. S6 in fact provides some glimpses on this question. What is in addition needed is a distance measurement between Walker A motif and ABC signature motif in both NBSs. If there is indeed asymmetric closure (a clear difference in these distances), that would be worth being mentioned. And if both NBSs are fully closed, this should be mentioned as well.

8) Line 307: "NBD2, which contains the catalytic nucleotide-binding site, performs a greater domain movement than NBD1, which hosts the deviant site."

The same is actually also seen in TM287/288. The authors may consider to cite:

<https://pubmed.ncbi.nlm.nih.gov/31113958/>

9) Line 312: "The single-site hydrolysis is a simplification of the symmetric ABC transporter system, whereby two ATP molecules are thought to complete the transport cycle⁴⁷".

It is not clear what is really meant with this statement. We guess the authors mean that hydrolysis of one ATP is sufficient to complete a transport cycle for Pdr5, whereas for homodimers (or ABCB1 with two consensus sites), two ATP are (seem to be?) needed.

10) Line 316: "Various studies report a high degree of structural similarity of the transition and the pre-hydrolytic states for other ABC transporters, indicating that ATP binding and not hydrolysis induce the conformational change in the transporter from the inward-facing to the outward-facing^{27,48–50}". In this context, one might add more citations on DEER studies where ATP-EDTA was shown to be sufficient to induce the inward-facing/outward-facing transition.

<https://pubs.acs.org/doi/abs/10.1021/jacs.7b12409>

<https://pubmed.ncbi.nlm.nih.gov/28051765/>

11) Line 332: "The Pdr5 exit channel might also contain the equivalent of the leucine gate found in ABCG2, which controls drug extrusion in that protein⁵¹ (Fig. S9)."

This statement would profit from functional/biochemical experiments. However, I would not consider them as essential (other than for R6G binding and the nucleotide-sensing motif).

Misspellings:

Line 83: domain is at the N-terminus of both pseudo-protomers, resulting in a reverse Fig. S2 legend: define abbrev. for "SM"

Line 246: "using Pdr5 samples with added ATP, indicating that that the majority of ATP was"

Line 320: "state" or "conformation" is missing at the end of the sentence.

COMMENTS TO THE REVIEWERS

Reviewer #1 (Remarks to the Author):

In this manuscript, Harris et al. present four cryo-EM structures of peptidisc-reconstituted Pdr5, *S. cerevisiae* ABC transporter, belonging to the pleiotropic drug resistance family. The authors provide a detailed description of those structures, emphasizing structural features, important for the transporter function. The structures, described here, are a valuable addition to the ABC transporter gallery, and the main findings, including asymmetric conformational changes and the substrate-binding mode, are of high interest to the wide scientific community. Cryo-EM analysis meets the current standards, however, there are many flaws in the representation and description of the data, which need to be fixed:

1. The summary lacks details – it summarizes the main findings, but does not even tell which method was used to get them. The authors' statement "structural and functional results reveal details of an ATP-driven conformational cycle" leaves me wondering what kind of functional results the authors are referring to. Actually, there is not a single experimental evidence for the biochemistry in this paper, which is largely unexpected. Authors should at least show ATPase activity assays to demonstrate that the transporter is active after reconstitution into peptidiscs.

We have changed the summary to reflect the content of the manuscript more appropriately. We have also included information about the purification of Pdr5 in detergent (Fig. S1) and data for the ATPase activity of Pdr5 reconstituted in peptidisc (Fig. S2). Pdr5 reconstituted in peptidisc indeed retains its ATPase activity. On the point of 'functional' results, we performed additional experiments and added data to test the role of the newly identified 'sensor' in the linker domain in context of the ATPase and substrate transport activities (Fig. S7 and S8).

Information on the hydrolysis speed would additionally support the findings that Pdr5 hydrolyzed most of ATP prior to plunge-freezing of EM grids.

ATPase activity in peptidisc revealed a V_{\max} value of 72.8 ± 2.9 nmol / (min*mg). Using this value and in the context of the experimental setup (20 μ L sample, 1.5 mg/mL Pdr5, 2 mM ATP, and incubation time of 20 min), we can reasonably expect almost complete hydrolysis.

Moreover, the statement in the summary about the coupling of nucleotide hydrolysis with drug efflux is not correct as Pdr5 is an uncoupled transporter, which authors also mention themselves in the text later.

We apologize for using infelicitous wording. We have rephrased this in the revised version of the manuscript.

2. For cryo-EM analysis, Pdr5 was reconstituted in peptidiscs, but the authors do not include any figure showing this. At least an SDS-PAGE gel, proving that reconstitution was successful, should be shown. In general, it is also recommended to include a SEC profile into supplementary figures to provide insights into sample homogeneity.

We have added a supplementary figure (Fig. S2) to address this point of the Reviewer.

3. On the second paragraph of page 4, the authors write: "The resolution of the maps ranged between 2.8 Å and 3.5 Å", but according to Table S2, the resolution of maps is between 2.9 and 3.8 Å. This should be fixed accordingly, and the deposition codes for the maps and models should be included into the manuscript.

We have changed the text to reflect the correct resolution range. The deposition codes are now included in the text.

4. As this is an entirely structure-based manuscript, it is necessary to provide maps and models for evaluation by the reviewers.

We happily share the maps and models if requested by the reviewers or editor. For now, we have included the PDB validation reports in the revised submission.

5. The authors do not see magnesium at the non-catalytic site in their structures. While this is very likely for the inward-facing structures, how sure are the authors about the absence of magnesium in the AOV-Pdr5 structure, which has the lowest resolution? I have some doubts about making such claims at 3.8 Å resolution. The authors should support these claims in the supplementary figures by demonstrating model and map overlay at the nucleotide-binding sites for all the structures.

We entirely agree and have added the caveat that the non-catalytic site of AOV-Pdr5 might not contain Mg because of the limited resolution. The map/model overlay suggested by the reviewer has now been included for clarity (Fig. 5).

6. There are still debates in the field about conformational changes following ATP binding and hydrolysis, therefore, simply dropping a statement like in the subtitle on page 8 (“ATP binding induces a change to outward-facing conformation of Pdr5”) without a solid proof is not a good idea. I am sure that this statement is correct, but it should not be used here as the data does not show this - the authors do not have ATP bound outward-facing structure; they only have ADP-vanadate structure, which is not the same.

We have changed the subtitle to read “Vanadate-trapped Pdr5 adopts outward-facing conformation” and changed the wording in this section to make it clearer that the interpretation is a hypothesis.

7. In the methods, the authors should write at which concentration Pdr5 was used for cryo-EM grid preparation.

We have added the information about Pdr5 concentration used to prepare the grids.

8. The authors mention that they used CHAPSO to improve particle distribution, however, they do not show the angular distribution plots. Such plots should be added to the supplementary data. I would also recommend adding the processing workflow to the supplementary figures.

We have added a new supplementary figure that illustrates the angular distribution of particles (Fig. S3c) and the processing workflow (Fig. S4).

9. The representation of the structures is very confusing. It is generally accepted to display cytoplasmic side on the bottom and the extracellular side on the top, but the authors display everything inverted. I am not sure if the authors wanted to show their creativity or to highlight the reverse transmembrane topology, but such representation simply brings confusion and, therefore, should be changed to promote direct comparability with other studies. There is enough confusion in the field regarding nomenclature, naming of the different states and other things, so it is really a bad idea to introduce yet another way of representation. A good figure should deliver the main message without the need of additional text, which is absolutely not the case in the figures of this manuscript. But I do like that the authors display the peptidisc.

We have inverted all the Pdr5 structure representations in the paper, according to the Reviewer's remark. The extracellular domain of Pdr5 is now on the top, and the cytoplasmic part points towards the bottom of the page.

10. The authors should be consistent with their naming – in panels E-F of Figure 1, the ADP-Pdr5 map is called ATP-Pdr5. At the end of the caption of the same figure, the authors write “Arrowheads point to selected amino acid residues”, but it is not clear why these amino acid residues were selected. In general, the authors should be more specific in figure captions, for example, in Fig. S1 it is written: “Figure panels illustrate some of the steps in the structural reconstruction of apo-Pdr5 (A–D), ADP-Pdr5 (E–F), AOV-Pdr5 (G–H), and R6G-Pdr5 (I–J).” This is too unspecific and also implies that the authors display the same steps for all four structures, but this is not the case here. I would suggest removing this sentence and

simplifying the caption by explaining which step and for which protein is shown on each panel.

We have corrected the label on Fig. 1 and improved the caption in Fig. S3 (revised Fig. S1).

11. Fig. 2 lacks consistency – it shows NBS1 and NBS2 for ADP-Pdr5, but only NBS2 for AOV-Pdr5 structure. The authors should also include NBS1 for AOV-Pdr5 here. It would be a good idea to show the densities for nucleotides and the substrate in this figure, but if it gets overcrowded, such densities with corresponding fits should be at least shown in supplementary figures as requested before (see comment 5).

Fig. 2 has been expanded to include NBS1 in the AOV-Pdr5 structure (Fig 2d).

Reviewer #2 (Remarks to the Author):

As a mammalian p-glycoprotein homolog in *Saccharomyces cerevisiae*, Pdr5 has attracted a lot of attention. This work elucidates the molecular structure of Pdr5 for the first time based on Cryo-EM and addresses a key problem in Pdr5 research. It will be very helpful for the further studies on structural and functional relation of Pdr5.

We thank the reviewer for this positive and encouraging statement.

Reviewer #3 (Remarks to the Author):

Harris et al. present here a nicely conducted structural study on the well-characterized and “famous” yeast ABC transporter Pdr5. Four cryo-EM structures are presented, three (apo, ADP-bound and ADDP+R6G-bound) in inward-facing states and one (vanadate trapped) in the outward-facing state. Together with the structural data on drug binding (Rhodamine-6G), this structure series allows to formulate a transport mechanism (which in fact agrees with transport mechanisms of type II ABC exporters postulated earlier).

While these are not the first structures of a type II ABC exporter, the work nevertheless provides highly important novel insights into structural peculiarities of Pdr5 (and other members of the pleiotropic drug resistance transporter subfamily), namely a nucleotide-sensing linker domain, a unique extracellular domain, highly asymmetric nucleotide binding sites and finally an extra N-terminal domain with unknown function.

The structural work appears mostly solid (see some mild criticism below) and the manuscript is well written and intelligible to readers outside of the ABC transporter community. The figures are informative, aesthetical and clear. Provided the authors generate at least some extra functional data (see suggestions below) and place the work better into the larger context of ABC transporters (again suggestions are listed below), I consider this manuscript as a very good match for Nature Communications.

We thank the reviewer for this positive and encouraging statement.

Major points

1) Given the structural similarities to ABCG2, Pdr5 (at least in this reviewer’s view) belongs as well to the same ABC transporter subclass, namely it is a type II ABC exporter (older nomenclature) or a type V ABC transporter (newer nomenclature). The manuscript in this form falls short in providing any information regarding the structural classification of Pdr5, which very likely will lead to confusion in the field (currently, the message comes across that Pdr5 is the founding member of a new ABC transporter subclass not structurally investigated thus far).

We have rephrased this section to make clear that Pdr5 is only the founding member of a sub-family of the ABCG family. We have also now included more description of the structural classification of Pdr5 and added a new figure showing a structural comparison with ABCG2, ABCG5/G8, TarGH, and Wmz-Wmt (Fig. 2). We have also

added a table that lists r.m.s.d. of structural alignments of Pdr5 (in both outward- and inward-facing conformations) with the above and other transporters.

2) Assuming that Pdr5 belongs to the type II ABC exporters, it is very important to highlight similarities and differences between the existing and these novel structures. A comprehensive list of structures can be found here <https://febs.onlinelibrary.wiley.com/doi/10.1002/1873-3468.13935>. It is worth noting that for ABCG2, both inward- and outward-facing structures had been determined, allowing for a very close examination of similarities and peculiarities. Further, it is worth to be noted that with the structure of ABCG5/8, a heterodimeric member of this class of transporters had been solved already once. A comparison to this ABCG5/8 structure (PDB: 5DO7) is certainly needed. Another interesting aspect is whether Pdr5 is related more closely to the eukaryotic members of the subfamily (ABCA1, ABCG2, ABCG5/8) or to the prokaryotic members of the subfamily (Wzm-WztN and TarGH).

We thank the reviewer for the comment and have added references to other structures in the revised text. Also see our comment above.

3) The most exciting structural finding of this study is (in this reviewer's view) the nucleotide-sensing linker domain. The authors pinpoint a highly conserved motif (MQKGEIL) being critically involved in mediating important contacts. The authors should confirm with functional and biochemical experiments that this motif really plays a role and it has to be investigated, whether and how this motif is involved in nucleotide sensing at the molecular level.

We have performed the requested experiments. We have created eight mutants within the linker region and analysed their drug resistance (Fig. S7 and Table S3) as well as ATPase activity (Fig. S8 and Table S4). The discussion of these new results has been incorporated in the main body of the revised version (section: "Pdr5 contains a nucleotide-sensing linker domain").

4) The authors claim that they observed R6G binding. However, they do not provide cryo-EM densities to further support this claim (in fact, a comparison of the structures determined with ADP versus ADP+R6G are needed to judge). As for point 3, the authors should generate a few mutants in the R6G binding region and characterize them at the functional/biochemical level to further substantiate their findings (as a positive control, the authors can carry along a previously described mutant, namely S1360F).

We have now included the cryo-EM density of the map with R6G bound to support the statement about binding (Fig. 3). We explored the binding pocket for substitutions but could not propose substitutions that would impact solely R6G binding without impacting conformational switching. However, we do discuss in the text the impact of mutations from the literature on transport, and these are consistent with the proposed mode of substrate interaction. Furthermore, we show that the R6G binding site of Pdr5 coincides with the substrate binding site in ABCG2 (Fig. S10d).

5) Line 366/367: "These structures now also provide the basis to explain important functional mutants of Pdr5 (for a selection see Table S1)." These previously described mutants have to be discussed more thoroughly in a separate section.

We have included a new section that discusses the mutants from Table S5 (revised Table S1), titled "Pdr5 structure can rationalise many observed mutational phenotypes", which is also summarised graphically in Fig. 8.

6) The manuscript essentially lacks a conclusion (other than explaining the transport mechanism). In this reviewer's view, a conclusion could contain the novelty of findings of Pdr5 in the context of existing knowledge about the type II ABC exporter structures and mechanisms.

We have expanded the text with a "Conclusions" section in the revised text.

7) It may sound a bit silly, but why do the authors depict all structures “upside down”? It is common sense in the ABC transporter field to show the structures with the NBDs down and not up. Did this happen by “accident” or is it the intention of the authors to make Pdr5 look like very special?

This has been corrected. We now show the “classic” view of ABC transporters.

Minor points

1) There are multiple errors in Table S1:

- “Q951Q”? □ Q951G

Corrected.

- Maybe explain a bit more why these different mutations modulate the function of Pdr5
- E244G was not found in publication 74
- S1360F was found in publication 76, but not in 77 in which they describe S1368A
- S1386A was not found in publication 78. In this publication they describe V656L, P596L, L1367F, and A670S.
- Please check again all citations

All above mentioned references have been double-checked and corrected.

2) Is the transporter still active when reconstituted into peptidiscs? Of course, we see the different states, so in theory it must be active. However, it is of interest whether the transporter is less/similar/more active compared to in vivo and other in vitro studies? The in vitro study (14) the authors are referring to was performed with detergent-purified Pdr5.

We confirmed that Pdr5 is active in peptidisc. Please see our response to comment 1 by Reviewer #1.

3) Line 268-270: “Because of this, we surmise that there is no cross-talk between the substrate binding pocket and the rest of the transporter in the inward-facing conformation, which is supported by the fact that Pdr5 is an uncoupled transporter.” However, there is ADP in the consensus ATP binding site when R6G is bound. From this, can one really assume that the binding pocket is not communicating with the rest of the transporter? In the referred publication (22), drug-mediated inhibition of Pdr5-specific ATPase activity was observed, based on which it was claimed the transporter is uncoupled.

We have removed the comment.

4) Line 233: is there a structural explanation, why magnesium was not found to be bound in the degenerate site?

Most likely, the residues in the site do not support the coordination of the magnesium ion, but we acknowledge in the revision that the resolution of the map is not high enough to establish, with certainty, the absence of the interaction. We have adjusted the wording of this section to reflect this uncertainty.

5) Line 245: “The ADP-Pdr5 structure was solved using Pdr5 samples with added ATP, indicating that the majority of ATP was hydrolysed before flash-freezing in liquid ethane.” Is this really plausible? How much ATP was added and how much Pdr5 was present? And what is the ATPase activity of Pdr5 in peptidisc at this temperature of incubation (see also point above). And how long was the transporter incubated before freezing? Based on these parameters, the authors can estimate how much ATP has been converted to ADP. We could imagine, that ATP turnover was only partial, but that ADP/ATP exchange at the canonical site might be slow?

We are reasonably certain that the hydrolysed nucleotide ADP is present in the catalytic site but not ATP. As cryo-EM provides an averaged 3D reconstruction of protein particles in the ice, in the case of partial ATP hydrolysis, we would expect to see a combination of the different nucleotide states. Please also see our response to comment 1 by Reviewer #1

6) Line 258: "Pdr5 can utilise other nucleotides to fuel its transport activity^{14,43}, and this can be rationalised by our structures from the distribution of nucleotide contacts." What exactly is meant with this? Which residues are likely of relevance for binding e.g. GTP versus ATP?

What we intended to convey here is that the structure of the pocket is such that it can, in principle, support the binding of other nucleotides, i.e., the nucleobase does not make any appreciable contacts with the residues of the protein. We have amended the sentence in the paper to reflect this better.

7) Line 296: "This occurs mainly through the interaction of the nucleotide, bound in NBS2, with the D-loop of the NBD1, and follows a pattern that is almost universally conserved amongst ABC transporters and other proteins that share a similar fold^{1,11}." Nucleotide sandwiching most likely has also occurred at the degenerate NBS1. At least in TmrAB, Mrp1 and TM287/288, both NBSs are completely closed. The way the authors describe the situation here, it implies that only the ATP at the consensus NBS2 glues the NBDs together. Fig. S6 in fact provides some glimpses on this question. What is in addition needed is a distance measurement between Walker A motif and ABC signature motif in both NBSs. If there is indeed asymmetric closure (a clear difference in these distances), that would be worth being mentioned. And if both NBSs are fully closed, this should be mentioned as well.

It is the case that both the NBS1 and NBS2 close around the nucleotide. We have added the suggested measurement in the text to quantify this movement.

8) Line 307: "NBD2, which contains the catalytic nucleotide-binding site, performs a greater domain movement than NBD1, which hosts the deviant site."

The same is actually also seen in TM287/288. The authors may consider to cite:

<https://pubmed.ncbi.nlm.nih.gov/31113958/>

We added the suggested reference to the manuscript.

9) Line 312: "The single-site hydrolysis is a simplification of the symmetric ABC transporter system, whereby two ATP molecules are thought to complete the transport cycle⁴⁷".

It is not clear what is really meant with this statement. We guess the authors mean that hydrolysis of one ATP is sufficient to complete a transport cycle for Pdr5, whereas for homodimers (or ABCB1 with two consensus sites), two ATP are (seem to be?) needed.

This is indeed what was meant by this statement. We have revised the original sentence to provide more clarity.

10) Line 316: "Various studies report a high degree of structural similarity of the transition and the pre-hydrolytic states for other ABC transporters, indicating that ATP binding and not hydrolysis induce the conformational change in the transporter from the inward-facing to the outward-facing^{27,48–50}".

In this context, one might add more citations on DEER studies where ATP-EDTA was shown to be sufficient to induce the inward-facing/outward-facing transition.

<https://pubs.acs.org/doi/abs/10.1021/jacs.7b12409>

<https://pubmed.ncbi.nlm.nih.gov/28051765/>

We added the suggested references to the manuscript.

11) Line 332: "The Pdr5 exit channel might also contain the equivalent of the leucine gate found in ABCG2, which controls drug extrusion in that protein⁵¹ (Fig. S9)."

This statement would profit from functional/biochemical experiments. However, I would not consider them as essential (other than for R6G binding and the nucleotide-sensing motif).

We have decided not to interrogate this putative gate further, as we feel it somewhat more distant from the focus of this paper (even further than the investigation of the binding pocket residues).

Misspellings:

Line 83: domain is at the N-terminus of both pseudo-protomers, resulting in a reverse

Corrected.

Fig. S2 legend: define abbrev. for "SM"

We have explained the abbreviation.

Line 246: "using Pdr5 samples with added ATP, indicating that that the majority of ATP was"

Corrected.

Line 320: "state" or "conformation" is missing at the end of the sentence.

Corrected.

REVIEWERS' COMMENTS

Reviewer #1 (Remarks to the Author):

The authors have addressed my concerns and significantly improved the manuscript. The study is well performed, reads well and delivers a story of high interest to the ABC transporter community, therefore, I can only recommend it for publication.

Reviewer #3 (Remarks to the Author):

The authors have addressed all my major concerns and the manuscript has substantially improved. The fact that the authors addressed the functional role of the MQKGEIL motif with further experiments is highly appreciated.
The manuscript can be published in this form.